# PIFE: Progressive Insight driven Feature Engineering via Multimodal Reasoning

## Abstract

Despite significant advances in Automated Machine Learning (AutoML), one of its persistent blind spots remains the automation of data-centric tasks such as exploratory data analysis (EDA), contextual insight extraction, and feature engineering. These steps-often more critical than model selection itself-are still largely manual, domain-specific, and reliant on human intuition. Existing automated feature engineering (AutoFE) techniques either rely on rigid transformation sets or complex optimization strategies that struggle with interpretability and fail to leverage the rich, visual cues that guide human decision-making. In this work, we introduce **PIFE**: **P**rogressive **I**nsight driven **F**eature **E**ngineering via Multimodal Reasoning; a novel AutoFE framework that employs multimodal language models as collaborative agents in an iterative pipeline. PIFE systematically performs automated EDA, generating statistical summaries and visualizations that are jointly interpreted through text–vision reasoning. These multimodal insights inform the synthesis of candidate transformations, represented as symbolic programs in executable Python code to ensure interpretability and reproducibility. By coupling iterative insight extraction with validation-driven refinement, PIFE produces high-quality, interpretable features that consistently enhance the performance of diverse predictive models, outperforming existing AutoFE baselines. Extensive experiments across diverse tabular datasets demonstrate the effectiveness and adaptability of our approach, paving the way for a new class of human-aligned, insight-aware AutoFE systems.

## 1 Introduction

The rapid evolution of automated machine learning (AutoML) has significantly advanced model selection, hyperparameter tuning, and performance optimization (Chopde et al., 2025; Aragão et al., 2025; Hutter et al., 2019; Feurer et al., 2015; Olson & Moore, 2016; Erickson et al., 2020). However, AutoML tools continue to face limitations in automating data engineering tasks, particularly exploratory data analysis (EDA), feature insight extraction, and systematic feature engineering. These data-centric activities often dominate real-world machine learning workflows, where the transformation of raw tabular data into meaningful representations is a bigger bottleneck than model fitting. Although automated feature engineering (AutoFE) has emerged as a subfield within AutoML, traditional methods, such as expansion-reduction algorithms (Kanter & Veeramachaneni, 2015; Lam et al., 2021; Kaul et al., 2017; Shi et al., 2020; Katz et al., 2016) typically construct large search spaces composed of manually defined transformation operations and employ various search or optimization strategies to identify effective features. However, these methods are often limited by the rigidity of their predefined operations and generally lack the integration of domain-specific knowledge (Zhang et al., 2023).

To reduce the cost of searching through large feature space and generate data-driven features, learning-based AutoFE methods are proposed (Khurana et al., 2018; Nargesian et al., 2017; Chen et al., 2019; Zhu et al., 2022). However, these methods fall short in incorporating domain expertise and contextual insights from data exploration. Similarly, evolutionary methods focus on optimization strategies but neglect the nuanced, often visual cues that inform human-driven feature creation. Language-powered systems like CAAFE (Hollmann et al., 2023) and LLM-FE (Abhyankar et al., 2025) have shown promise in bridging this gap by generating candidate features based on dataset context and iterative refinement. However, these methods remain limited by feature simplicity, a

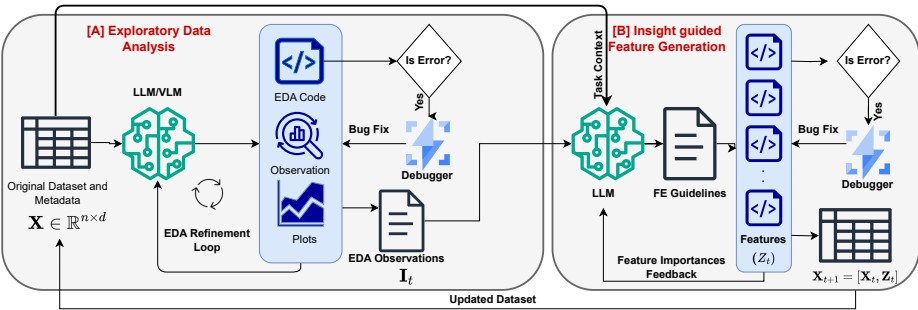

Figure 1: Overview of PIFE Framework. For a given dataset, PIFE goes through the following steps. (a) Exploratory Data Analysis and Data Insight Generation(b) Feature Generation via Symbolic Program Synthesis

lack of interpretability regarding why certain features should be created (as opposed to merely explaining what they represent), and the absence of a truly data-driven approach. Furthermore, visual patterns-such as distributional anomalies, multivariate correlations, or interaction structures-remain underutilized despite their centrality in manual feature engineering workflows.

This gap highlights the opportunity to harness recent advancements in Large Language Models that understand both textual and visual modalities to build a framework for automated insight extraction and feature engineering. These models are capable of interpreting not only data descriptions but also visualizations such as histograms, scatter plots, and heatmaps; elements that humans frequently rely on during feature engineering in real-world scenarios. Yet, their potential in automating feature generation grounded in rich exploratory insights has remained largely unexplored. A more effective AutoFE pipeline must seamlessly incorporate insights from both narrative and visual representations of data.

To address these challenges, we propose a novel AutoFE framework that integrates iterative EDA cycles using a unified reasoning engine capable of understanding both text and plots. Our system performs repeated rounds of insight extraction to build a deeper and comprehensive contextual understanding of the dataset, which then guides feature generation. The generated candidate features are evaluated using a downstream predictive model, where the corresponding feature importance serves as feedback to subsequent feature generation cycles. We argue that the broader process of feature engineering can be naturally decomposed into two complementary stages: (i) feature generation and (ii) feature selection. While the former aims to enrich the feature space, the latter plays a critical role in filtering redundant or irrelevant features and selecting an optimal subset that maximizes task performance. To emphasize the importance of this selection step, we conduct extensive experiments comparing diverse feature selection methodologies and demonstrate that incorporating effective selection strategies can further enhance the performance of automated feature engineering (AutoFE) pipelines. This feedback-driven, context-rich process enhances automation and interpretability, while aligning closely with the iterative and insight-informed nature of human data science workflows.

**Contributions.** The key contributions of this work are as follows:

- We propose the first automated feature engineering framework that integrates textual and visual exploratory data insights into a unified, iterative pipeline.

- We highlight the central role of feature selection in AutoFE by conducting extensive experiments across diverse selection methodologies, showing that effective selection strategies further boost both predictive performance and interpretability compared to state-of-the-art AutoFE methods.

- We conduct extensive experiments across various tabular datasets, demonstrating superior performance and enhanced interpretability compared to state-of-the-art AutoFE methods.

## 2 RELATED WORK

Automated feature engineering (AutoFE) has emerged as a critical component in simplifying the model development pipeline by transforming raw data into informative representations. Early efforts such as Deep Feature Synthesis (DFS) (Kanter & Veeramachaneni, 2015), LFE (Nargesian et al., 2017), Cognito (Khurana et al., 2016), AutoFeat (Horn et al., 2020), and OpenFE (Zhang et al., 2023) employed exhaustive enumeration or heuristic-based transformation strategies, often relying on predefined operator sets and lacking semantic understanding of domain-specific relationships. OpenFE extended traditional methods via an expansion-reduction framework with incremental feature boosting and pruning, achieving strong empirical performance but still limited by its lack of contextual reasoning and domain adaptivity.

More recent methods address these limitations through learning-based strategies. TransGraph (Khurana et al., 2018), Neural Feature Search (NFS) (Chen et al., 2019), and DIFER (Zhu et al., 2022) adopted reinforcement learning and differentiable architecture search to explore high-dimensional transformation spaces more efficiently. DIFER, in particular, proposed a differentiable encoder-predictor-decoder pipeline to optimize feature embeddings in continuous space, though it primarily supports numerical transformations. FETCH (Li et al., 2023) approached AutoFE as a Markov Decision Process, using a policy network trained across datasets to learn transferable feature construction policies. Despite its generalizability, FETCH suffers from sparse rewards and computational overhead, echoing challenges seen in DIFER and NFS.

In parallel, large language models (LLMs) have shown promise in data-centric applications, leveraging their contextual understanding to perform data wrangling, imputation, and semantic reasoning over tabular data (Hegselmann et al., 2023; Narayan et al., 2022; Vos et al., 2022). CAAFE (Hollmann et al., 2023) was among the first to explore LLM-driven feature engineering, generating features based on dataset metadata and producing human-readable descriptions. However, it lacks iterative feedback from prior search histories and relies on column descriptions to create features. OCTree (Nam et al., 2024) augmented this by incorporating decision-tree reasoning into LLM prompts, offering structured, contextual feedback for subsequent feature generation. OCTree performs iterative refinement of feature generation rules until improvements in downstream performance. While effective, this approach is susceptible to poor initialization in LLMs, which can hinder convergence and overall effectiveness.

LLM-FE (Abhyankar et al., 2025) takes a different approach by casting feature engineering as a program synthesis problem. It combines LLM reasoning, evolutionary strategies, and memory buffers to maintain a population of candidate features, using both validation scores and information-theoretic feedback to guide selection. This method addresses sparse rewards and brittleness in prompting; it introduces a new challenge: by conditioning future generations on previously successful feature transformations, the model may become biased toward certain transformation patterns. This can skew the process toward exploitation, limiting its ability to explore novel and potentially better features.

Our framework leverages LLMs as agents for in-depth EDA, identifying outliers, feature interactions, and distributional patterns, which are mapped to candidate transformations and validated via downstream performance-balancing automation, interpretability, and domain adaptivity beyond prior AutoFE methods.

> ### A. Rationale for EDA
>
> To study interaction and non-linear effects, I examine the joint surface of `Average_Covered_Charges` and `reimbursement_rate` against the target to detect curvature and interaction; cross-category heterogeneity via a DRG-by-State heatmap to expose geographic–clinical interplay; and utilization effects by relating `Total_Discharges` to the target with a smooth trend plus binned means to reveal non-linear scaling and heteroskedasticity. Expected outputs highlight regimes where combinations of drivers jointly shift payment levels, and where certain DRG–State combinations systematically differ. Limitation: heavy tails may distort binning; quantile-based bins and downsampling are applied as needed.

**B. EDA Code Snippet**

```python
bins_x = quantile_bins(df_surface[num_x], q=12)
bins_y = quantile_bins(df_surface[num_y], q=10)
df_surface = df_surface.assign(_bx=bins_x, _by=bins_y)
# Aggregate target mean for each 2D bin
surf_agg = (df_surface.groupby(['_bx', '_by'])[target_col]
                .mean().unstack('_by').sort_index())
```

**C. Generated Plots**

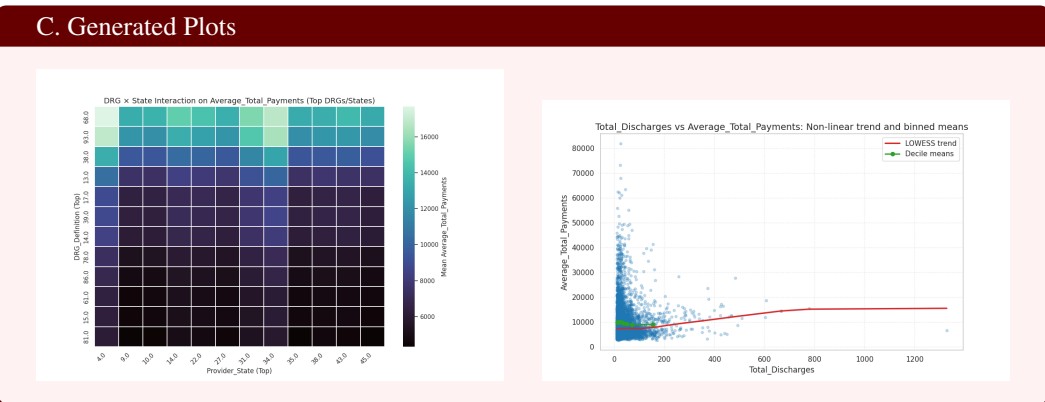

**D. EDA Analysis**

**Total_Discharges vs target scatter with LOWESS:** The relationship is non-linear with a concave (increasing-then-flattening) trend; variance is high at low volumes and shrinks as discharges grow. This motivates monotone, variance-stabilizing transforms and regime features (e.g., ranks/bins).

**DRG × State heatmap:** There is strong cross-category heterogeneity—within the same DRG, `Provider_State` causes sizable shifts in mean payments, and DRGs also differ markedly in their baseline level. This supports both main effects (`DRG`, `State`) and their interaction.

Figure 2: Exemplery run of PIFE on `medical_charges_nominal` dataset showing the process of generating data insights. First, the rationale is generated, creating a plan for exploratory analysis to be conducted. In B, this plan is translated into a program for EDA. In C, when the code is executed, analysis plots are generated, and at the end, plots and statistics are analyzed to generate statistics.

## 3 METHODOLOGY

PIFE is an iterative, insight-driven feature engineering framework that employs Large Language Models (LLMs) and Vision-Language Models (VLMs) to emulate the reasoning workflow of a data scientist. Instead of generating transformations directly from raw data, PIFE first constructs a multi-stage hierarchy of EDA insights-statistical and visual—and then synthesizes symbolic transformation programs guided by these insights. Each iteration integrates downstream feedback, allowing the system to refine subsequent EDA and transformation strategies. Figure **??** provides an overview of the complete pipeline.

We divide our methodology into three parts: (1) problem definition and iterative objective, (2) hierarchical EDA insight extraction using LLMs and VLMs, and (3) transformation rule synthesis and downstream refinement. This provides additional explanation before introducing Algorithm 1.

### 3.1 PROBLEM FORMULATION

Let $\mathbf{X} \in \mathbb{R}^{n \times d}$ denote an input tabular dataset with $n$ samples and $d$ features, and let $\mathbf{y} \in \mathbb{R}^n$ represent the target variable. The goal of PIFE is to produce an enriched feature set $\mathbf{X}^* \in \mathbb{R}^{n \times d^*}$,

---

**Algorithm 1** PIFE: Insight-Driven Iterative Feature Engineering

---

**Input:** Dataset $\mathbf{X}$, target $\mathbf{y}$, prior context $\mathcal{P}$
**Parameters:** EDA rounds $K$, iterations $T_f$, max features $N$
**Output:** Feature sets $\{\mathbf{Z}_t\}_{t=1}^{T_f}$

1: Initialize $t \leftarrow 0$, $\mathbf{X}_0 \leftarrow \mathbf{X}$
2: **while** $t < T_f$ **do**
3:     Construct dataset description $\mathcal{D}_t$        {column names, types}
4:     $\mathcal{C}_t^{(0)} \leftarrow \mathcal{D}_t \cup \mathcal{P}$        {initial LLM context}
5:     **for** $k = 1$ to $K$ **do**
6:        $S_t^{(k)} \leftarrow \text{ComputeStats}(\mathbf{X}_t)$        {quantiles, skewness, correlations}
7:        $\mathbf{I}_t^{(k)} \leftarrow \text{LLM}(\mathcal{C}_t^{(k-1)} \cup S_t^{(k)})$        {statistical insights}
8:        $\mathcal{V}_t^{(k)} \leftarrow \text{Visualize}(\mathbf{X}_t)$        {plots and distributions}
9:        $\mathbf{I}_{\text{vlm}}^{(k)} \leftarrow \text{VLM}(\mathcal{V}_t^{(k)})$        {visual insights}
10:        $\mathcal{C}_t^{(k)} \leftarrow \mathcal{C}_t^{(k-1)} \cup \mathbf{I}_t^{(k)} \cup \mathbf{I}_{\text{vlm}}^{(k)}$        {expand reasoning context}
11:     **end for**
12:     $\mathbf{I}_t \leftarrow \bigcup_{k=1}^{K} \left( \mathbf{I}_t^{(k)} \cup \mathbf{I}_{\text{vlm}}^{(k)} \right)$        {consolidated insights}
13:     $\mathcal{G}_t \leftarrow \text{LLM}(\mathbf{I}_t)$        {transformation guidelines}
14:     $\mathbf{Z}_t \leftarrow \text{GenerateFeatures}(\mathcal{G}_t, N)$        {symbolic programs $\rightarrow$ features}
15:     $\hat{\mathbf{y}}_t \leftarrow \mathcal{M}(\mathbf{X}_t \cup \mathbf{Z}_t)$        {evaluate features}
16:     $\phi_t \leftarrow \text{Importance}(\mathcal{M}, \mathbf{Z}_t)$        {feature importances}
17:     Update feedback: $\mathcal{P} \leftarrow \mathcal{P} \cup \phi_t$
18:     Update dataset: $\mathbf{X}_{t+1} \leftarrow [\mathbf{X}_t, \mathbf{Z}_t]$
19:     $t \leftarrow t + 1$
20: **end while**
    =0

---

where $d^* \geq d$, by repeatedly generating candidate transformations and incorporating only those that improve predictive performance.

At each iteration $t$, the framework generates a set of candidate features $\mathbf{Z}_t$ via a two-stage process: hierarchical EDA insight extraction and symbolic transformation synthesis. These features are appended to the current dataset and evaluated using a downstream model $\mathcal{M}$. Model predictions, along with feature importances $\phi_t$, form a feedback context $\mathcal{P}$ that guides subsequent EDA rounds:

$$\hat{\mathbf{y}}_t = \mathcal{M}(\mathbf{X}_t \cup \mathbf{Z}_t), \quad \phi_t = \text{Importance}(\mathcal{M}, \mathbf{Z}_t), \quad \mathcal{P} \leftarrow \mathcal{P} \cup \phi_t.$$

The dataset is updated as $\mathbf{X}_{t+1} = [\mathbf{X}_t, \mathbf{Z}_t]$. After $T_f$ iterations, the best-performing feature set is selected:

$$\mathbf{X}^* = \arg\max_{\mathbf{X}_t} \text{CV\_Score}(\mathcal{M}, \mathbf{X}_t, \mathbf{y}).$$

### 3.2 HIERARCHICAL EDA INSIGHT EXTRACTION

A central aspect of PIFE is its hierarchical EDA structure, which mitigates randomness by guiding the LLM through increasingly sophisticated analyses. This structure ensures that early iterations capture coarse patterns, while later ones explore deeper interactions.

At iteration $t$, PIFE constructs the dataset description:

$$\mathcal{D}_t = \{(c_i, \tau_i)\}_{i=1}^{d_t},$$

where $c_i$ is the feature name and $\tau_i$ its type (numerical, categorical, temporal). This is merged with accumulated feedback context $\mathcal{P}$ to initialize reasoning.

For $k = 1, \ldots, K$, PIFE performs structured EDA:

- **Stage 0:** univariate distributions, summary statistics, skewness, missingness.

- **Stage 1:** correlations, pairwise relationships, non-linear dependencies.

- **Stage 2:** temporal effects, categorical interactions, and outlier structure.

At each stage, the LLM receives statistical summaries, while the VLM analyzes visualizations—e.g., density plots, bivariate scatter plots, grouped temporal charts. Consolidating statistical and visual signals produces the insight set $\mathbf{I}_t$ upon which transformation rules are constructed.

### 3.3 Symbolic Transformation Program Generation

Given the consolidated insight set $\mathbf{I}_t$, an LLM produces a collection of transformation guidelines $\mathcal{G}_t$, each describing a candidate operation. These guidelines are synthesized into executable **Python** programs which are run to produce the actual feature columns. Concretely, the system generates a set of Python feature-generation scripts, executes them, and collects the resulting candidate transformations:

$$\mathbf{Z}_t = \{z_{t,1}, \ldots, z_{t,n_t}\}, \qquad n_t \leq N,$$

where each $z_{t,i}$ is materialized by executing the corresponding Python program.

We use Reverse Polish Notation (RPN) only as an auxiliary representation for *analysis* and validation: RPN helps inspect operator ordering, detect redundant or ill-formed expressions, and perform lightweight static checks on transformation pipelines before execution. This dual strategy-readable, executable Python for generation and RPN for program-order analysis-preserves interpretability, facilitates debugging, and avoids opaque, black-box feature construction.

### 3.4 Iterative Refinement and Downstream Feedback

Finally, each iteration evaluates the generated features using a downstream model $\mathcal{M}$. Feature importances $\phi_t$ provide a compact summary of which transformations contributed meaningfully. These importances are reintegrated into the feedback context $\mathcal{P}$, enabling the next iteration to focus on relevant transformations, avoid redundant patterns, and maintain diversity of features.
This closed-loop design-EDA $\rightarrow$ synthesis $\rightarrow$ evaluation; ensures that PIFE incrementally refines the search space and maintains interpretability while improving predictive performance.

## 4 Experiments

In this section, we evaluate PIFE over several classification and regression datasets spanning across various domains such as healthcare, finance, real estate, weather forecasting, etc. Our experiments reveal that PIFE consistently improves the performance of predictive models (Section 4.2). Ablation studies (Section 4.3) show that data-grounded insight extraction helps create features that are more aligned to the downstream objective. Also, feature selection is often not focused on in the scope of feature engineering, which plays a pivotal role in boosting the performance of predictive models.

### 4.1 Experimental Setup

**Datasets.** We evaluate PIFE across 22 tabular tasks, encompassing both classification and regression objectives. The majority of datasets are drawn from prior AutoFE literature (Li et al., 2023), ensuring coverage of diverse domains, scales, and complexity levels. Additionally, we include a set of recent Kaggle datasets (Kaggle) (e.g., ps5_episode_3, ps5_episode_4), which were not part of the pretraining corpora of large language models, providing a test of robustness to novel data sources. Detailed dataset information is provided in Table 6.

**Metrics.** For classification tasks, we use F1-micro (Sokolova & Lapalme, 2009), and for regression tasks, we use $(1 - \text{relative absolute error})$ (Shcherbakov et al., 2013) as the evaluation metric for downstream models. Higher values correspond to better model performance. To quantify improvements, we also report the percentage increase over a baseline score, reflecting relative efficacy.

**Baselines.** We compare PIFE against a diverse set of baseline methods representing key paradigms in automated feature engineering, all of which have publicly available, executable open-source implementations to ensure reproducibility. Heuristic-based approaches include AutoFeat Horn et al. (2020), DFS (Deep Feature Synthesis) Kanter & Veeramachaneni (2015), and OpenFE Zhang et al. (2023), which rely on expansion and reduction strategies over predefined transformations. Among LLM-based approaches, we include CAAFE Hollmann et al. (2023), leveraging LLMs for feature generation and refinement using metadata, prompts, or reasoning frameworks, and OCTree Nam et al. (2024), which employs rule-based feature generation and CART decision tree inputs to improve feature quality. The CAAFE and OCTree implementations were adapted to support newer LLM models and extended to handle both classification and regression tasks, with additional metrics introduced for fairer comparison. However, certain recent methods, such as LLM-FE Abhyankar et al. (2025), are excluded due to incomplete publication of

methodology and evaluation details, which would limit fair comparison. Additional discussion is provided in Appendix A.11.

**Implementation Details.** To ensure reliable evaluation, we perform 5-fold cross-validation on the training set, mitigating overfitting and yielding robust performance estimates. Results are reported as mean $\pm$ standard deviation over three random seeds (42, 44, 46) to account for stochasticity in LLMs and training pipelines. We use the given specific versions of LLMs and VLMs: gpt-4.1-2025-04-14 and gpt-5-2025-08-07. Given that gpt-4.1 and gpt-5 include built-in vision capabilities, we adopt them as our core VLM components for all experimental evaluations. For fairness, all datasets are preprocessed by imputing or removing missing values and encoding categorical variables, as most downstream models lack native support. Further details on LLM prompting strategies (Appendix A.13), hyperparameters (Appendix A.10), and additional configuration settings are provided in Appendix A.

Table 1: Comparison of AutoFE methods across method compatibility and performance (mean $\pm$ std) for different LLMs. Tick (✓) indicates presence, cross (✗) indicates absence of a feature. Results are averaged across 3 seeds, with each seed evaluated using 5-fold cross-validation and Random Forest as the predictive model. We report the f1-micro score for classification and (1-rae) for regression datasets.

| Method | Context Aware | Without Description | Interpretable Feature | LLM | Avg. Score (%) |
|--------|:---:|:---:|:---:|:---:|:---:|
| Baseline | ✗ | ✗ | ✗ | - | $0.7558 \pm 0.1017$ |
| DFS | ✗ | ✓ | ✗ | - | $0.7718 \pm 0.1101$ (2.12%) |
| Autofeat | ✗ | ✓ | ✗ | - | $0.7651 \pm 0.0948$ (1.23%) |
| OpenFE | ✗ | ✓ | ✗ | - | $0.7684 \pm 0.0922$ (1.67%) |
| CAAFE | ✓ | ✗ | ✓ | gpt-4.1[*] gpt-5[*] | $0.7791 \pm 0.1068$ (3.08%) $0.7900 \pm 0.0996$ (4.53%) |
| OCTree | ✗ | ✓ | ✗ | gpt-4.1[*] gpt-5[*] | $0.7745 \pm 0.0958$ (2.47%) $0.7750 \pm 0.0969$ (2.54%) |
| PIFE (Ours) | ✓ | ✓ | ✓ | gpt-4.1[*] gpt-5[*] | **$0.7908 \pm 0.1105$ (4.63%)** **$0.7917 \pm 0.1037$ (4.75%)** |

[*] All experiments were conducted using LLM versions gpt-4.1-2025-04-14 and gpt-5-2025-08-07.

## 4.2 PERFORMANCE COMPARISONS

Table 1 highlights PIFE as the most effective and practical AutoFE method: it attains the top average score while preserving semantic interpretability and context awareness, and it works even without dataset descriptions. Classical baselines (DFS, Autofeat, OpenFE) offer modest gains but lack contextual understanding and interpretability. Among LLM-based methods, PIFE leads fairly: with gpt-4.1, it improves 4.63% over the Baseline (without feature engineering) versus 3.08% for CAAFE; with gpt-5, the gap narrows, but PIFE still edges ahead, suggesting stronger reasoning models reduce, but do not erase, method-level differences. These results demonstrate that PiFE consistently improves performance across seeds and folds, producing interpretable, context-aware features with minimal dependence on the latest LLMs. Full results are reported in Appendix Table 7.

## 4.3 ABLATIONS

**Impact of EDA.** To assess the contribution of the EDA component in insight-driven feature generation, we compare model performance with and without EDA in Table 2. Even without EDA, the generated features are optimized and achieve competitive results. However, EDA provides a data-grounded mechanism for feature generation, enabling the capture of complex relationships and trends that are difficult to model when relying solely on LLM optimization or metric-based feedback. We observe that certain datasets exhibit limited performance gains from inclusion of EDA.

Table 2: Performance comparison of Baseline, w/o EDA, and w/ EDA across multiple datasets under the No Feature Selection setting. The best value per dataset is highlighted in bold. Percentage improvement over the baseline is shown in parentheses. We report f1-micro score for classification (*) and (1-relative absolute error) for regression (†) datasets.

| Dataset | Baseline | w/o EDA | w/ EDA |
|---|---|---|---|
| adult* | 0.850 | 0.852 (0.3%) | **0.853 (0.3%)** |
| fertility* | 0.829 | 0.873 (5.4%) | **0.880 (6.2%)** |
| medical_charges_nominal† | 0.891 | **0.922 (3.5%)** | 0.903 (1.4%) |
| openml_586† | 0.613 | 0.729 (18.8%) | **0.772 (25.9%)** |
| pima_indian* | 0.700 | **0.759 (8.3%)** | 0.754 (7.7%) |
| ps5_episode_4† | 0.571 | 0.577 (1.2%) | **0.578 (1.3%)** |

**Interpretability of Features from EDA.** From our observations above, across a wide range of datasets spanning diverse domains, the overall performance improvements remain modest, a closer examination of what our framework actually achieves is essential for understanding the value of incorporating visual cues from EDA into the feature engineering process.

Our experimentation runs reveals interesting insights on how EDA-driven visual insights lead to more grounded, interpretable, and context-aware feature construction, in contrast to older approaches that rely heavily on randomness or brute-force search. By examining a few representative examples (Appendix A.6), we can better illustrate the specific benefits, the interpretability gains, and the underlying reasoning behind each feature generated. These examples highlight how visual patterns-often invisible to purely statistical or automated methods-guide the system toward features that meaningfully capture complex relationships within the data.

**Feature Selection: Trade-offs and Downstream Performance.** Although PIFE can generate interpretable and statistically strong features, they might not reflect the same way on performance. Therefore, effective feature selection is crucial to identify the optimal subset of features. To evaluate this, we compare three approaches: Model-based Feature Importance(MFI), Conditional Mutual Information-based Bayesian Optimization (CMI-BO), and Genetic Algorithm(GA) (see the Appendix A.12). CMI-BO and MBFI did not show any improvement in performance. The genetic algorithm, while computationally intensive, gave the best performance compared to others. We can see that the set of features generated by PIFE enabled good exploration inthe Genetic Algorithm. In this way, PIFE complements these approaches by producing a concise, high-quality subset of features, making downstream optimization more efficient while maintaining strong predictive performance.

Table 3: Performance comparison of PIFE with feature selection during run (using CMI and validation) and after the run (using a Genetic Algorithm), with GPT-5 as the downstream model. Reported metrics are F1-micro for classification datasets and (1-relative absolute error) for regression datasets.

| Dataset | Baseline | PIFE | PIFE(CMI-BO) | PIFE(MFI) | PIFE(GA) |
|---|---|---|---|---|---|
| adult | 0.851±0.001 | 0.853±0.001 | 0.853±0.001 | 0.851±0.002 | **0.855±0.001** |
| fertility | 0.853±0.006 | 0.88±0.010 | 0.873±0.015 | 0.873±0.006 | **0.89±0.008** |
| openml_586 | 0.662±0.009 | 0.765±0.012 | 0.742±0.025 | 0.760±0.001 | **0.792±0.002** |
| pima_indian | 0.739±0.010 | 0.754±0.01 | 0.755±0.005 | 0.76±0.010 | **0.777±0.002** |

Feature transferability is critically dependent on model inductive bias. Transformations that encode tree-like, thresholding behaviour generally transfer well to tree ensembles but can degrade performance for neural or transformer architectures that favour smooth, continuous representations or learned embeddings. Moreover, near-ceiling baseline performance leaves little headroom for gains. Finally, transfer success is dataset-dependent, being modulated by sample size, noise, feature types, and the specific nature of engineered transformations. Engineered features should be validated on the intended downstream model family.

Table 4: Feature Transferability of PiFE-generated Features to Deep Learning Models (MLP, TabPFN(Hollmann et al., 2023), and HyperFast (Bonet et al., 2024)). * denotes classification and †denotes regression tasks. HyperFast (NA) only runs on classification tasks.

| Dataset | MLP | | TabPFN | | HyperFast | |
|---|---|---|---|---|---|---|
| | Baseline | PIFE (Ours) | Baseline | PIFE (Ours) | Baseline | PIFE (Ours) |
| hepatitis* | **0.862**±**0.020** | 0.852±0.016 | **0.832**±**0.005** | 0.826±0.005 | 0.815±0.015 | **0.843**±**0.007** |
| airfoil† | 0.735±0.001 | **0.802**±**0.001** | **0.886**±**0.002** | 0.884±0.001 | NA | NA |
| credit_approval* | **0.888**±**0.002** | 0.884±0.002 | **0.874**±**0.005** | 0.863±0.005 | **0.861**±**0.006** | 0.847±0.003 |
| spectf* | **0.828**±**0.003** | 0.81±0.014 | 0.798±0.005 | **0.806**±**0.005** | 0.792±0.007 | **0.805**±**0.01** |
| megawatt_1* | 0.900±0.008 | **0.908**±**0.004** | 0.893±0.003 | **0.895**±**0.008** | 0.865±0.008 | **0.884**±**0.007** |
| housing_boston† | 0.701±0.001 | **0.706**±**0.001** | **0.735**±**0.001** | 0.735±0.001 | NA | NA |

**Integrating with other AutoFE Methods.** Engineered features from PIFE can serve as input to other AutoFE frameworks. We experimented with OpenFE as the integrated framework and report the results in Table 5. Overall, integrating PIFE features with OpenFE didn't result in a substantial improvement. This can be seen as a positive outcome: PIFE already identifies a strong set of features on its own. By leveraging insights from exploratory data analysis (EDA) and domain knowledge encoded in LLMs, along with natural language descriptions of the data, PIFE generates features that are both meaningful and predictive. Even after exploring a large space of additional candidate features (∼2000) in OpenFE, there is little to no gain, and in some cases, performance slightly decreases. This highlights the robustness and quality of the features generated by PIFE.

**Feature Order Analysis.** PIFE is capable of creating higher-order meaningful features and can be scaled well for creating complex interactions between features due to its iterative nature of feature generation. We can see that some of the new features created have an order greater than the number of feature engineering steps. From this, we can infer that LLM is attempting to create complex, higher-order features in a single feature engineering step.

Table 5: Performance comparison of PIFE and PIFE† (extended to OpenFE) across competitions. Values: mean ± standard deviation. All results are based on gpt-5. f1-micro score for classification and (1-relative absolute error) for regression datasets.

| Competition | PIFE | PIFE† |
|---|---|---|
| adult | 0.851 ± 0.002 | **0.855 ± 0.002** |
| fertility | 0.870 ± 0.040 | 0.870 ± 0.040 |
| medical_charges_nominal | 0.905 ± 0.000 | **0.907 ± 0.001** |
| openml_586 | 0.773 ± 0.023 | **0.790 ± 0.010** |
| openml_607 | 0.732 ± 0.012 | **0.752 ± 0.023** |
| ps5_episode_4 | 0.578 ± 0.001 | **0.579 ± 0.016** |
| **Average** | 0.778 ± 0.107 | **0.780 ± 0.105** |

**Flexibility Towards Predictive Models.** PIFE generalizes well across various downstream predictive models, showing consistent improvements with Logistic Regression, Random Forest, and XGBoost. As expected, tree-based models, which can capture non-linear relationships among features, generally outperform linear models such as Logistic Regression.

## 5 CONCLUSION

We present PIFE, a multimodal AutoFE framework that leverages textual and visual insights from datasets to iteratively generate and select predictive features. By combining exploratory data analysis with LLM-guided reasoning, PIFE automates feature engineering in a way that mirrors human workflows, enhancing both performance and interpretability.

Our experiments across diverse tabular datasets demonstrate that effective feature selection amplifies the benefits of automated feature generation. However, PIFE has limitations: large feature sets can lead to prompt size constraints, automatically derived dataset descriptions and visualizations may be noisy, and LLMs can produce plausible but ungrounded features. Additionally, results can vary across random seeds, underscoring the importance of multiple runs for robustness.

Future work includes improving multimodal reasoning, fine-tuning models for better feature

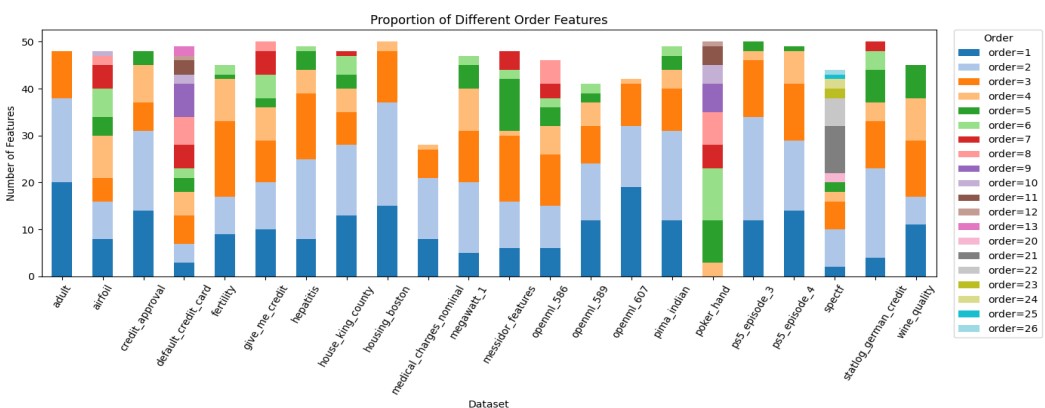

Figure 3: Order of features created per competition. This is based on the gpt-5 runs from Table 1

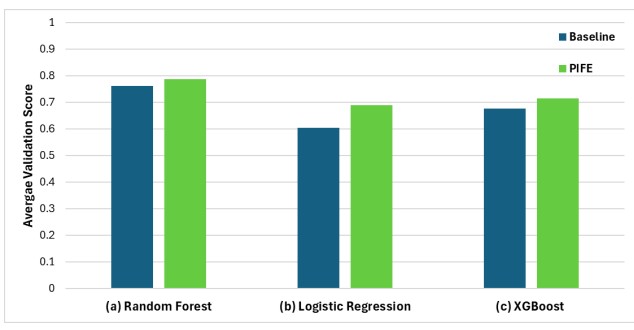

Figure 4: Performance of PIFE with predictive models. All results use GPT-5 as base LLM with the same hyperparameters as the main table. We report f1-micro for classification and (1-relative absolute error) for regression datasets.

generation, integrating human-in-the-loop interactions, and incorporating continuously updated datasets to reduce memorization biases. PIFE represents a step toward context-aware, interpretable, and robust automated feature engineering.

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

# A APPENDIX

## A.1 REPRODUCIBILITY

We release our code to ensure reproducibility of experiments at `https://anonymous.4open.science/r/pife-7FE5`. The repository includes the main PiFE pipeline, integrations with baseline frameworks, and scripts for ablation studies. Additionally, we provide the datasets used in our experiments, including modifications made to publicly available datasets to facilitate consistent evaluation.

## A.2 DATASET COLLECTION, PREPROCESSING AND RESULTS

The datasets are collected by ensuring that contamination is minimal. The formatted data description contains the task, dataset description, target, and objective. We are handling missing values by replacing them with numeric values using the mean and filling missing categorical values with the most frequent value. Categorical columns are encoded with ordinal encoding. The EDA process is informed about the preprocessing steps taken by providing the preprocessing context to the module. $C$ and $R$ in Table 6 represent Classification and Regression task types, respectively.

Table 6: Summary of Benchmark Datasets

| Name | Alias | Source | Type (C/R) | Inst./Feat. | Subject Area |
|------|-------|--------|-----------|-------------|--------------|
| Adult | adult | UCI | C | 48842 / 14 | Social Science |
| Credit Approval | credit_approval | UCI | C | 690 / 15 | Business |
| Default Credit Card | default_credit_card | UCI | C | 30000 / 23 | Business |
| Fertility | fertility | UCI | C | 100 / 9 | Health and Medicine |
| Give Me Some Credit | give_me_credit | Kaggle | C | 251503 / 10 | Finance |
| Hepatitis | hepatitis | UCI | C | 155 / 19 | Health and Medicine |
| Megawatt1 | megawatt_1 | OpenML | C | 253 / 37 | Mathematics |
| Diabetic Retinopathy Debrecen | messidor_features | UCI | C | 1151 / 19 | Health and Medicine |
| PimaIndian | pima_indian | Kaggle | C | 768 / 9 | Health and Medicine |
| Poker Hand | poker_hand | UCI | C | 1025010 / 10 | Games |
| Rainfall Dataset | ps5_episode_3 | Kaggle | C | 2920 / 11 | Weather and Climate |
| SPECTF | spectf | UCI | C | 267 / 44 | Health and Medicine |
| Statlog German Credit | statlog_german_credit | UCI | C | 1000 / 20 | Social Science |
| Wine Quality | wine_quality | UCI | C | 4898 / 11 | Business |
| Airfoil | airfoil | UCI | R | 1503 / 5 | Physics and Chemistry |
| House King County | house_king_county | Kaggle | R | 21613 / 20 | Business |
| Housing Boston | housing_boston | UCI | R | 506 / 13 | Finance |
| Medical Charges Nominal | medical_charges_nominal | OpenML | R | 163065 / 11 | Business |
| OpenML 586 | openml_586 | OpenML | R | 1000 / 25 | Mathematics |
| OpenML 589 | openml_589 | OpenML | R | 1000 / 25 | Mathematics |
| OpenML 607 | openml_607 | OpenML | R | 1000 / 50 | Mathematics |
| Podcast Listening Time | ps5_episode_4 | Kaggle | R | 1000000 / 10 | Entertainment |

Table 7: Performance comparison across different AutoFE methods. Values represent mean ± standard deviation of the metric score. We use F1-micro for classification and (1 - rae) for regression tasks. Random Forest was used as a downstream predictive model.

| Competition | Baseline (NO FE) | OpenFE | DFS | Autofeat | CAAFE gpt-4.1 | CAAFE gpt-5 | OCTREE gpt-4.1 | OCTREE gpt-5 | PIFE (Ours) gpt-4.1 | PIFE (Ours) gpt-5 |
|-------------|---------|--------|-----|----------|---------------|-------------|----------------|--------------|---------------------|-------------------|
| adult | 0.8511 ± 0.0007 | 0.8538 ± 0.0004 | 0.8547 ± 0.0009 | 0.8472 ± 0.0003 | 0.8532 ± 0.0021 | 0.8500 ± 0.0012 | 0.8547 ± 0.0008 | 0.8529 ± 0.0009 | **0.8853 ± 0.0609** | 0.8526 ± 0.0013 |
| airfoil | 0.7436 ± 0.0018 | 0.7473 ± 0.0097 | 0.7384 ± 0.0037 | 0.7508 ± 0.0042 | 0.7428 ± 0.0013 | **0.7613 ± 0.0024** | 0.7555 ± 0.0062 | 0.7513 ± 0.0045 | 0.7480 ± 0.0047 | 0.7588 ± 0.0060 |
| credit_approval | 0.8531 ± 0.0093 | 0.8585 ± 0.0051 | 0.8589 ± 0.0068 | 0.8604 ± 0.0080 | 0.8609 ± 0.0088 | 0.8565 ± 0.0124 | 0.8667 ± 0.0025 | 0.8681 ± 0.0063 | **0.8763 ± 0.0344** | 0.8659 ± 0.0010 |
| default_credit_card | 0.8070 ± 0.0007 | 0.8072 ± 0.0007 | 0.8079 ± 0.0008 | 0.8087 ± 0.0021 | 0.8061 ± 0.0015 | 0.8067 ± 0.0013 | **0.8093 ± 0.0009** | 0.8090 ± 0.0002 | 0.8092 ± 0.0008 | 0.8088 ± 0.0003 |
| fertility | 0.8533 ± 0.0058 | 0.8633 ± 0.0115 | 0.8733 ± 0.0058 | 0.8533 ± 0.0058 | 0.8600 ± 0.0100 | 0.8533 ± 0.0058 | **0.8900 ± 0.0000** | 0.8900 ± 0.0100 | 0.8733 ± 0.0153 | 0.8800 ± 0.0100 |
| give_me_credit | 0.9336 ± 0.0001 | 0.9328 ± 0.0002 | 0.9334 ± 0.0003 | 0.9342 ± 0.0004 | 0.9339 ± 0.0005 | 0.9335 ± 0.0002 | 0.9341 ± 0.0001 | **0.9342 ± 0.0004** | 0.9337 ± 0.0001 | 0.9337 ± 0.0002 |
| hepatitis | 0.8151 ± 0.0372 | 0.8172 ± 0.0244 | 0.7871 ± 0.0171 | 0.8194 ± 0.0171 | 0.8280 ± 0.0037 | 0.8086 ± 0.0074 | 0.8366 ± 0.0099 | **0.8473 ± 0.0207** | 0.8151 ± 0.0134 | 0.8172 ± 0.0325 |
| house_king_county | 0.6865 ± 0.0015 | 0.6887 ± 0.0016 | 0.6792 ± 0.0011 | 0.6875 ± 0.0018 | 0.6878 ± 0.0037 | 0.6948 ± 0.0034 | 0.6883 ± 0.0008 | 0.6897 ± 0.0013 | **0.6958 ± 0.0010** | 0.6909 ± 0.0014 |
| housing_boston | 0.6388 ± 0.0063 | 0.6488 ± 0.0108 | 0.6306 ± 0.0024 | 0.6482 ± 0.0033 | 0.6380 ± 0.0100 | 0.6459 ± 0.0100 | **0.6547 ± 0.0037** | 0.6483 ± 0.0050 | 0.6422 ± 0.0044 | 0.6445 ± 0.0069 |
| medical_charges_nominal | 0.8926 ± 0.0002 | 0.8986 ± 0.0002 | 0.8914 ± 0.0003 | 0.8922 ± 0.0001 | 0.8954 ± 0.0021 | 0.8978 ± 0.0011 | 0.8929 ± 0.0002 | 0.8928 ± 0.0001 | 0.8991 ± 0.0007 | **0.9033 ± 0.0019** |
| megawatt_1 | 0.8920 ± 0.0099 | 0.8855 ± 0.0068 | 0.8802 ± 0.0098 | 0.8907 ± 0.0100 | 0.8920 ± 0.0099 | 0.8854 ± 0.0104 | 0.8973 ± 0.0099 | 0.8947 ± 0.0099 | 0.9000 ± 0.0022 | **0.9039 ± 0.0099** |
| messidor_features | 0.6539 ± 0.0128 | 0.7197 ± 0.0203 | 0.7219 ± 0.0092 | **0.7416 ± 0.0070** | 0.6733 ± 0.0066 | 0.6794 ± 0.0083 | 0.6814 ± 0.0018 | 0.6652 ± 0.0010 | 0.6858 ± 0.0196 | 0.6979 ± 0.0071 |
| openml_586 | 0.6619 ± 0.0093 | 0.7162 ± 0.0103 | 0.6666 ± 0.0134 | 0.7108 ± 0.0105 | 0.6880 ± 0.0296 | **0.7735 ± 0.0270** | 0.7200 ± 0.0033 | 0.7200 ± 0.0035 | 0.7185 ± 0.0309 | 0.7721 ± 0.0013 |
| openml_589 | 0.6557 ± 0.0032 | 0.7022 ± 0.0059 | 0.6750 ± 0.0031 | 0.6870 ± 0.0012 | 0.7208 ± 0.0068 | **0.7711 ± 0.0162** | 0.6961 ± 0.0036 | 0.6961 ± 0.0036 | 0.6954 ± 0.0341 | 0.7351 ± 0.0148 |
| openml_607 | 0.6362 ± 0.0083 | 0.7036 ± 0.0046 | 0.6326 ± 0.0092 | 0.6506 ± 0.0126 | 0.6986 ± 0.0535 | **0.7655 ± 0.0081** | 0.6916 ± 0.0048 | 0.6916 ± 0.0084 | 0.7225 ± 0.0147 | 0.7254 ± 0.0176 |
| pima_indian | 0.7391 ± 0.0007 | 0.7574 ± 0.0033 | 0.7452 ± 0.0111 | 0.7353 ± 0.0133 | 0.7404 ± 0.0041 | 0.7405 ± 0.0074 | 0.7587 ± 0.0138 | 0.7505 ± 0.0033 | **0.7609 ± 0.0087** | 0.7543 ± 0.0098 |
| poker_hand | 0.6862 ± 0.0041 | 0.6862 ± 0.0041 | 0.9973 ± 0.0001 | 0.6862 ± 0.0041 | **1.0000 ± 0.0000** | **1.0000 ± 0.0000** | 0.7370 ± 0.0365 | 0.7856 ± 0.0579 | 0.9965 ± 0.0039 | **1.0000 ± 0.0000** |
| ps5_episode_3 | 0.8511 ± 0.0036 | 0.8490 ± 0.0028 | 0.8486 ± 0.0045 | 0.8511 ± 0.0036 | 0.8551 ± 0.0013 | 0.8560 ± 0.0018 | 0.8601 ± 0.0025 | 0.8565 ± 0.0056 | **0.8696 ± 0.0107** | 0.8554 ± 0.0019 |
| ps5_episode_4 | 0.5767 ± 0.0008 | 0.5791 ± 0.0049 | 0.5736 ± 0.0000 | 0.5767 ± 0.0003 | 0.5771 ± 0.0001 | 0.5773 ± 0.0005 | 0.5772 ± 0.0001 | 0.5775 ± 0.0002 | 0.5775 ± 0.0002 | **0.5784 ± 0.0011** |
| spectf | 0.7926 ± 0.0057 | 0.7851 ± 0.0265 | 0.7826 ± 0.0247 | 0.7926 ± 0.0057 | 0.7851 ± 0.0021 | 0.8114 ± 0.0056 | 0.8200 ± 0.0100 | 0.8064 ± 0.0076 | **0.8764 ± 0.1007** | 0.8278 ± 0.0210 |
| statlog_german_credit | 0.7507 ± 0.0038 | 0.7443 ± 0.0080 | 0.7457 ± 0.0087 | 0.7523 ± 0.0108 | 0.7490 ± 0.0052 | 0.7533 ± 0.0047 | 0.7547 ± 0.0038 | 0.7597 ± 0.0025 | 0.7587 ± 0.0045 | **0.7683 ± 0.0061** |
| wine_quality | 0.6575 ± 0.0057 | 0.6600 ± 0.0073 | 0.6545 ± 0.0055 | 0.6564 ± 0.0047 | 0.6545 ± 0.0047 | 0.6575 ± 0.0057 | **0.6623 ± 0.0028** | 0.6619 ± 0.0024 | 0.6572 ± 0.0017 | 0.6608 ± 0.0024 |
| **Average Score** | 0.7558 ± 0.1017 | 0.7684 ± 0.0922 | 0.7718 ± 0.1101 | 0.7651 ± 0.0948 | 0.7791 ± 0.1068 | 0.7900 ± 0.0996 | 0.7745 ± 0.0958 | 0.7750 ± 0.0969 | 0.7908 ± 0.1105 | **0.7917 ± 0.1037** |

## A.3 Evaluating PIFE Under Realistic Train-Test Splits

In our setup, we sought to closely mimic how feature engineering is typically performed in real-world applications by practitioners. Since a central objective of machine learning is to generalize effectively to unseen data, we designed our evaluation of PiFE to reflect this scenario. Each dataset was randomly split into training and test sets in a 70/30 ratio. The feature search process was restricted to the training split, where cross-validation was applied, while the test split was kept strictly unseen and used only for final evaluation. To mitigate order-related bias, all datasets were shuffled with fixed random seeds prior to splitting. This procedure may introduce slight variations in the reported scores compared to earlier tables. The detailed results of this evaluation are presented in Table 8.

Table 8: Performance comparison across competitions. Values are mean $\pm$ std (tiny). Best values in each Train/Test column group are in bold.

| Competition | Baseline | | CAAFE | | OpenFE | | OCTree | | PIFE | |
|---|---|---|---|---|---|---|---|---|---|---|
| | Train | Test | Train | Test | Train | Test | Train | Test | Train | Test |
| adult | 0.8493 ±0.0006 | 0.8495 ±0.0014 | 0.8504 ±0.0007 | 0.8487 ±0.0044 | **0.8530 ±0.0003** | 0.8420 ±0.0067 | 0.8526 ±0.0018 | **0.8518 ±0.0033** | 0.8504 ±0.0011 | 0.8517 ±0.0030 |
| fertility | 0.8571 ±0.0286 | **0.9000 ±0.0333** | 0.8619 ±0.0360 | 0.8667 ±0.0333 | 0.8571 ±0.0247 | 0.8667 ±0.0000 | 0.8905 ±0.0218 | 0.8556 ±0.0192 | **0.9095 ±0.0360** | 0.8667 ±0.0000 |
| medical_charges_nominal | 0.8913 ±0.0006 | 0.8918 ±0.0004 | **0.9001 ±0.0069** | 0.8978 ±0.0049 | 0.8975 ±0.0005 | 0.8782 ±0.0058 | 0.8916 ±0.0004 | 0.8919 ±0.0002 | 0.9000 ±0.0011 | **0.8982 ±0.0027** |
| openml_586 | 0.6221 ±0.0158 | 0.6603 ±0.0195 | **0.7713 ±0.0260** | **0.7461 ±0.0291** | 0.6780 ±0.0327 | 0.6992 ±0.0066 | 0.6927 ±0.0147 | 0.7133 ±0.0178 | 0.7294 ±0.0170 | 0.7271 ±0.0292 |
| pima_indian | 0.7344 ±0.0336 | 0.7287 ±0.0265 | 0.7455 ±0.0244 | 0.7359 ±0.0263 | 0.7524 ±0.0195 | 0.7524 ±0.0222 | 0.7692 ±0.0153 | 0.7273 ±0.0338 | **0.7698 ±0.0173** | **0.7677 ±0.0275** |
| ps5_episode_4 | 0.5714 ±0.0006 | 0.5753 ±0.0008 | 0.5738 ±0.0027 | **0.5772 ±0.0030** | **0.5744 ±0.0002** | 0.5557 ±0.0005 | 0.5721 ±0.0001 | 0.5757 ±0.0006 | 0.5733 ±0.0004 | 0.5703 ±0.0039 |
| **Average** | 0.7543 ±0.1268 | 0.7676 ±0.1268 | 0.7838 ±0.1123 | 0.7787 ±0.1128 | 0.7688 ±0.1182 | 0.7646 ±0.1173 | 0.7781 ±0.1199 | 0.7693 ±0.1134 | **0.7887 ±0.1205** | **0.7803 ±0.1148** |

The results indicate that PIFE consistently delivers strong performance, achieving both high cross-validation scores on the training data and correspondingly high accuracy on the test data across a large majority of datasets. This alignment suggests that effective feature engineering plans discovered by PIFE are not only tuned for the training split but also generalize reliably to unseen data, reinforcing the robustness of the approach.

## A.4 Additional Experiments and EDA-Guided Feature Examples for Time Series Data

This appendix provides (i) extended time-series benchmarks, (ii) illustrative EDA visualizations used by PIFE during feature synthesis, and (iii) detailed examples of derived features and their evaluation metadata. These additions complement the main paper and shows the robustness across modalities and the grounding of generated features in data-driven diagnostics.

### A.4.1 Time-Series Classification Results

To assess generalization beyond tabular data, we evaluated PIFE on five datasets from the UCR Time Series Archive. Using `gpt-5` as both LLM and VLM, PIFE consistently improves over strong baselines, demonstrating that EDA-driven feature synthesis transfers effectively to temporal domains. Table 9 reports accuracy and relative improvements.

Table 9: PIFE performance on UCR time-series classification datasets.

| Dataset | Baseline | PIFE | % Improvement |
|---|---|---|---|
| ItalyPowerDemand | $0.67 \pm 0.06$ | $0.70 \pm 0.05$ | 4.48 |
| GunPoint | $0.99 \pm 0.01$ | $1.00 \pm 0.00$ | 1.01 |
| Coffee | $0.83 \pm 0.02$ | $0.87 \pm 0.02$ | 4.82 |
| ECG200 | $0.96 \pm 0.01$ | $0.99 \pm 0.00$ | 3.13 |
| Beef | $0.82 \pm 0.01$ | $0.93 \pm 0.02$ | 13.41 |

These results show that PIFE's EDA routines; including autocorrelation profiling, summary lag statistics, seasonality checks, and rolling-window diagnostics; enable meaningful temporal feature construction without requiring modality-specific prompts or architectural changes.

### A.4.2 Illustrative EDA Visualizations and Derived Features

We present representative examples from two datasets (`coffee`, `ECG200`), demonstrating how PIFE leverages VLM-based interpretation of plots to guide feature design.

**Case Study: coffee** Figure 5 shows a VLM-interpreted 2D KDE scatter of $(t_{231}, t_{160})$. Listing 1 shows the interpretation of VLM, Listing 2 demonstrates the generated Python code for the interpreted feature, and Listing 3 shows evaluation metadata for the same feature.

Listing 1: VLM Interpretation for 2D KDE plot from coffee dataset

```
The global Pearson correlation is moderately negative ($-0.593$), while
    within-class correlations are weakly positive or near zero. This
    suggests that cross-band contrasts; particularly normalized
    differences; may highlight discriminative spectral shifts.
```

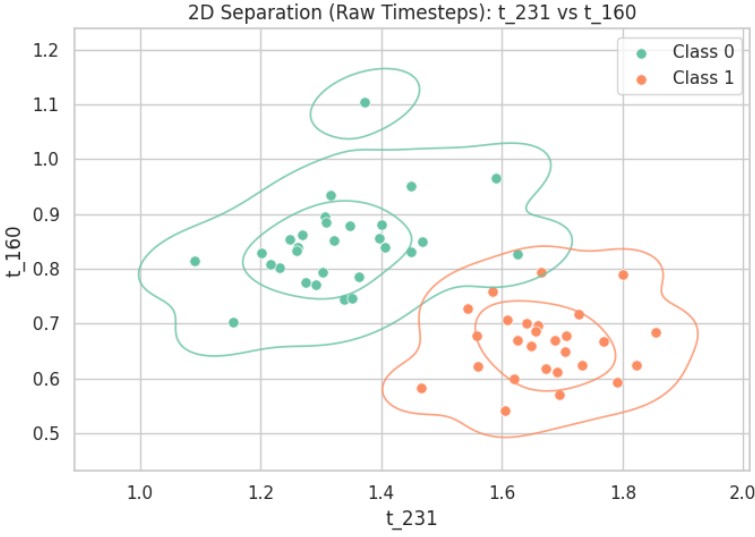

Figure 5: 2D KDE scatter for coffee (channels $t_{231}$ vs. $t_{160}$).

Listing 2: Cross-band normalized contrast feature (ASCII minus) from coffee dataset

```python
# Transformation 3: Cross-band normalized contrast on strongest late/mid
    pair -> 1 final feature
# crossband_normdiff_231_160 = (t_231 - t_160) / (t_231 + t_160)
den_train = df_train['t_231'] + df_train['t_160']
df_train['crossband_normdiff_231_160'] = np.where(
    np.abs(den_train) > EPS,
    (df_train['t_231'] - df_train['t_160']) / den_train,
    0.0
)
```

Listing 3: Evaluation metadata for crossband_normdiff_231_160 from coffee dataset

```json
{
  "name": "crossband_normdiff_231_160",
  "description": "Cross-band normalized difference contrasting late
      intensity at 231 against mid trough at 160.",
  "RPN": "t_231 t_160 - t_231 t_160 + /",
  "simplified_RPN": "t_231 t_160 - t_231 t_160 + /",
  "feature_order": 2,
  "transformation_order": 2,
  "order_feature_set": [
      "t_231",
      "t_160"
  ],
  "feature_importance_score": [0.1],
  "derived": true,
```

```
"status": "accepted",
"data_type": "float64",
"nature": "continuous",
"stats": {
    "n_missing": 0,
    "n_unique": 56,
    "mean": 0.32354085842005037,
    "std": 0.11405837153659522
}
}
```

**Case Study: ECG200**

Figure 6 visualizes class-specific trough and rebound timing. Listings 4, 5 and 6 shows VLM interpretation of Figure 6, python code for proposed feature and evaluation metadata for the feature.

Listing 4: VLM Interpretation of Temporal Events from ECG200 dataset

```
Trough timing is tightly localized ($t_{26}$--$t_{30}$) across classes,
    whereas rebound timing exhibits a clear class shift (early: $t_{45}$
    --$t_{47}$$; late: $t_{51}$--$t_{52}$).The inter-event spacing $(\text
    {rebound} - \text{trough})$ shows stable IQR with near-zero outliers,
     making window-based aggregation highly robust.
```

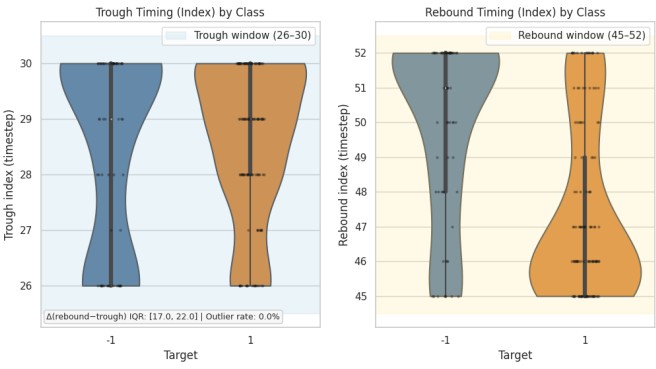

Figure 6: Trough and rebound timing patterns in `ECG200`.

Listing 5: Feature Proposition: Phase-Weighted Contrast from ECG200 dataset

```
# phase_weighted_contrast = (rebound_window_max    trough_min_26_30) *
    sigmoid(late    early)
trough_min_26_30 = df[['t_26', 't_27', 't_28', 't_29', 't_30']].min(axis
    =1)
rebound_window_max = pd.concat([early_reb_max_45_47, late_reb_max_50_52],
    axis=1).max(axis=1)
df['phase_weighted_contrast'] = (rebound_window_max - trough_min_26_30) *
    _sigmoid(reb_timing_bias)
```

Listing 6: Feature Evaluation: Phase-Weighted Contrast from ECG200 dataset

```
{
    "name": "phase_weighted_contrast",
    "description": "Core contrast (window max rebound minus trough min 26
        30 ) weighted by the sigmoid of  l a t e early  bias to focus on
        class-dependent phase differences.",
    "RPN": "t_50 t_51 max t_52 max t_45 t_46 max t_47 max max t_26 t_27
        min t_28 min t_29 t_30 min min    t_50 t_51 max t_52 max t_45
        t_46 max t_47 max    sigmoid   ",
```

```
    "simplified_RPN": "t_50 t_51 max t_52 max t_45 t_46 max t_47 max max
        t_26 t_27 min t_28 min t_29 t_30 min min    t_50 t_51 max t_52
        max t_45 t_46 max t_47 max    sigmoid   ",
    "feature_order": 11,
    "transformation_order": 3,
    "order_feature_set": [
        "t_45", "t_52", "t_30", "t_51", "t_26",
        "t_47", "t_50", "t_46", "t_27", "t_29", "t_28"
    ],
    "feature_importance_score": [0.06421524765915851],
    "derived": true,
    "status": "accepted",
    "data_type": "float64",
    "nature": "continuous",
    "stats": {
        "n_missing": 0,
        "n_unique": 200,
        "mean": 1.1571704660492668,
        "std": 0.4176547980018742
    }
}
```

### A.4.3 DISCUSSION: DEPENDENCE ON EDA DIVERSITY

PIFE is intentionally modular: extending to time-series involved editing only a small number of functions while keeping prompts unchanged. However, fully unstructured domains such as graphs require fundamentally different transformations and inductive biases, and thus fall outside the scope of this work. We note that datasets with limited structure or strong existing features may show smaller gains, whereas the provided time-series experiments highlight cases with more substantial improvement.

## A.5 HALLUCINATION MITIGATION IN PIFE

A potential concern when using Large Language Models (LLMs) and Vision Language Models (VLMs) for automated feature engineering is the risk of hallucinated or ungrounded features. PIFE incorporates several mechanisms to ensure generated features are grounded to underlying data.

### A.5.1 GROUNDING FEATURE GENERATION IN EDA

PIFE constrains the LLM to operate primarily on insights derived from Exploratory Data Analysis (EDA), while also leveraging its broader domain knowledge where relevant. Rather than producing features based purely on memorized patterns or unsupported priors, the system requires the LLM to justify every proposed transformation using:

- statistical summaries (distributional differences, correlations, heterogeneity), and
- visual evidence (scatter patterns, KDE structure, phase shifts, temporal signatures).

This grounding ensures that features reflect actual data-dependent structure, reducing the likelihood of hallucinations that conflict with observed statistics or visual trends.

### A.5.2 RELIABILITY OF VLM INTERPRETATIONS

Although VLMs can occasionally misinterpret plots, we can quantify and improve their reliability by benchmarking them on chart-understanding datasets such as **ChartQA** Masry et al. (2022) and **ChartQA-Pro** Masry et al. (2025). These benchmarks evaluate whether a model correctly extracts semantics, trends, and quantitative information from charts, offering a principled way to assess and compare VLM robustness.

Incorporating periodic evaluations on such benchmarks allows future extensions of PIFE to monitor VLM reliability across plot types, identify systematic failure modes (e.g., reading scales, interpreting trends), and select or fine-tune VLMs better suited for EDA-driven tasks.

### A.5.3 DOWNSTREAM VALIDATION AS A SAFETY LAYER

To further reduce the impact of potential hallucinations, PIFE performs downstream checks including:

- cross-validation on generated features,

- feature-importance pruning,

- comparison against baseline models,

- rejection of unstable or degenerately correlated features.

These mechanisms collectively form a practical defense against spurious or ungrounded features, ensuring that only empirically validated features are carried forward.

Overall, while hallucination remains a challenge for LLM-VLM systems, PIFE's design grounding in EDA, VLM benchmarking, and downstream validation; provides a robust and empirically supported mitigation strategy.

### A.6 EDA-DRIVEN INTERPRETABILITY: DATASET-SPECIFIC AGENT TRAJECTORIES EXAMPLES

This section presents qualitative trajectories generated by PIFE during EDA–Feature Engineering cycles. For each example dataset, we visualize key patterns, summarize LLM/VLM-derived insights, and list the corresponding feature transformations proposed by the agent.

---

**Example 1: `credit_approval`**

**Seed:** 44
**Baseline:** 0.86377
**PIFE (GPT-4.1):** 0.915942
**Other Frameworks:**

- OCTree (GPT-5): 0.87536
- CAAFE (GPT-5): 0.86812

---

**EDA 1: Distribution of A2**

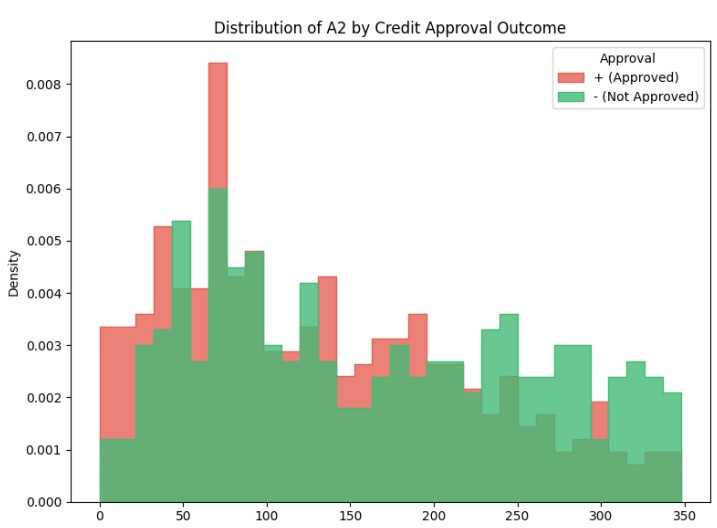

**Insight:** The distribution of `A2` across the target classes shows a clear shift: *Not Approved* applicants tend to have higher `A2` values, while *Approved* applicants are concentrated at lower values. Although the classes overlap, very high `A2` values are more frequent among the negative class, suggesting non-linear or threshold effects.

**Feature Suggestion.**

- **Feature:** `A2`

- **Transformations:** `log`, `sqrt_abs`

- **Rationale:** The right-skewed distribution and visible class separation indicate that compressing outliers (via log/sqrt) improves model sensitivity to threshold behaviors.

---

**EDA 2: Behavior of `A2_div_A8_minmax`**

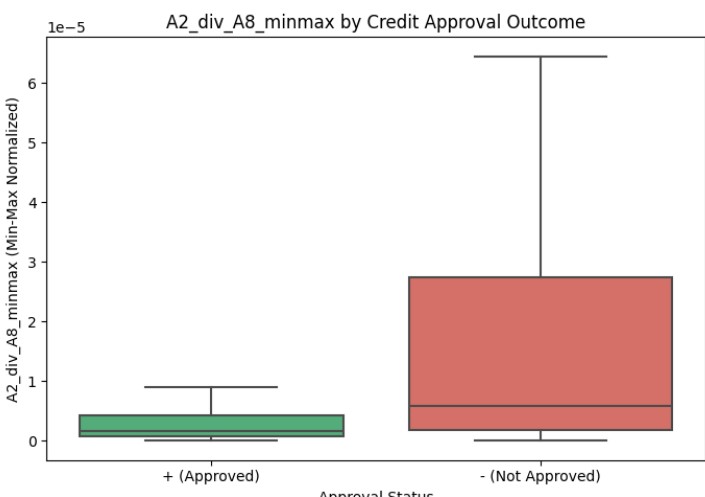

**Insight:** A boxplot of the normalized ratio `A2_div_A8_minmax` reveals that *Not Approved* applicants show a higher median and increased spread. The interquartile range and upper whisker are substantially larger for the negative class, indicating that extreme/high values are discriminative.

---

**Feature Suggestion.**

- **Features:** `A2`, `A8`

- **Transformations:**

  - Ratio: `A2 / A8`
  - `min_max` normalization
  - Decile or quintile binning

- **Rationale:** The ratio is strongly discriminative; binning emphasizes non-linear and tail behaviors correlated with rejection risk.

**Outlier-Aware Feature.**

- **Feature:** `A2_div_A8_minmax`

- **Transformation:** Outlier binning (bottom 5%, middle, top 5%) to generate a categorical indicator.

- **Rationale:** The presence of extreme values has high predictive value; explicit outlier signaling helps the model capture rare high-risk profiles.

---

**Example 2:** `spectf`

**Seed:** 44
**Baseline:** 0.79378
**PIFE (GPT-5):** 0.84647
**Other Frameworks:**

- CAAFE (GPT-5): 0.81621
- OCTree (GPT-5): 0.81251

---

**EDA 1: Pairwise Synergy (MI-based)**

Top Pairwise Interaction Synergy with Diagnosis
MI(Xi,Xj;y) − max(MI(Xi;y), MI(Xj;y))

**Insight:** The mutual-information synergy plot highlights several strong nonlinear joint effects:

- The strongest interaction is between `F21S` and `WorstRelDef_top4`, indicating that stress in ROI-21 interacts with maximal relative deficits across key ROIs.
- High synergy is observed for `worst_deficit` paired with `F21S` or `F13S`, suggesting that regional deficits amplify discriminative stress signals.
- Global burden features (e.g., `ischemic_burden_sum` and its `sqrt` variant) combined with `MinShare_top3` capture profiles exhibiting both high global ischemia and local weaknesses.
- Interactions among ROI stress features (e.g., `F20S × F13S`) provide information beyond individual ROIs.

---

**Feature Suggestion.**

- **Feature:** `min_max(F21S) × min_max(WorstRelDef_top4)`
- **Rationale:** Normalized multiplication captures the strongest observed synergy while mitigating scale effects.

**EDA 2: Plots**

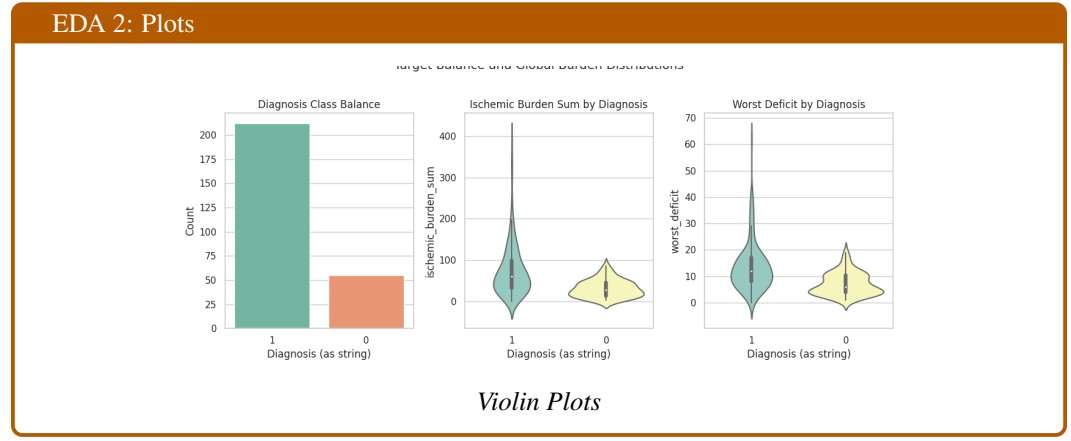

*Violin Plots*

**EDA 2: Plots**

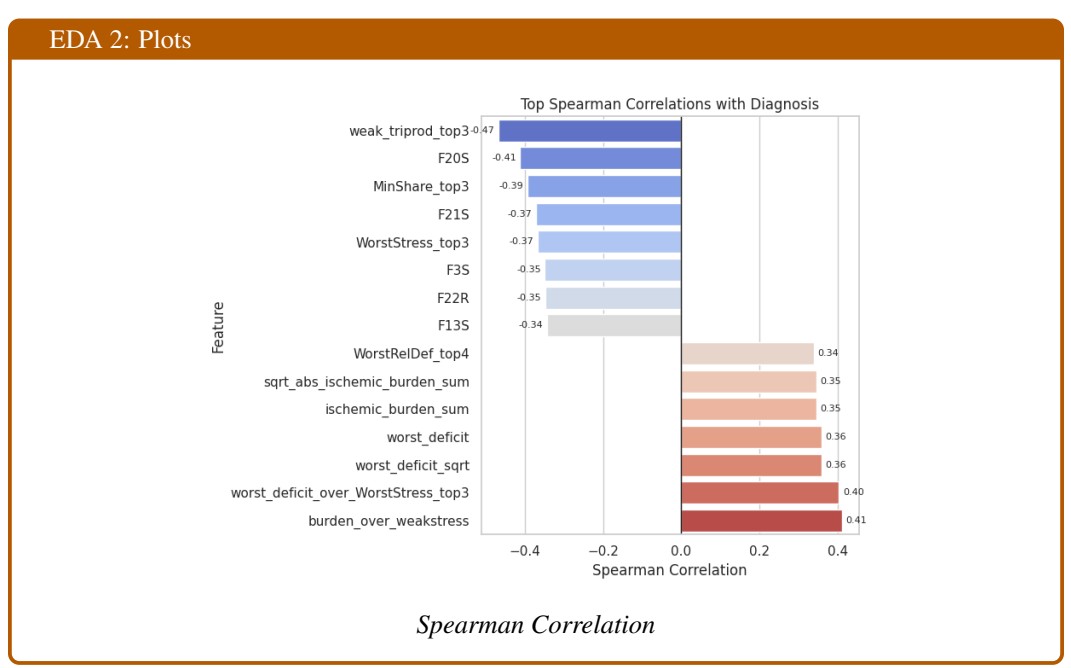

*Spearman Correlation*

**EDA 2: Plots**

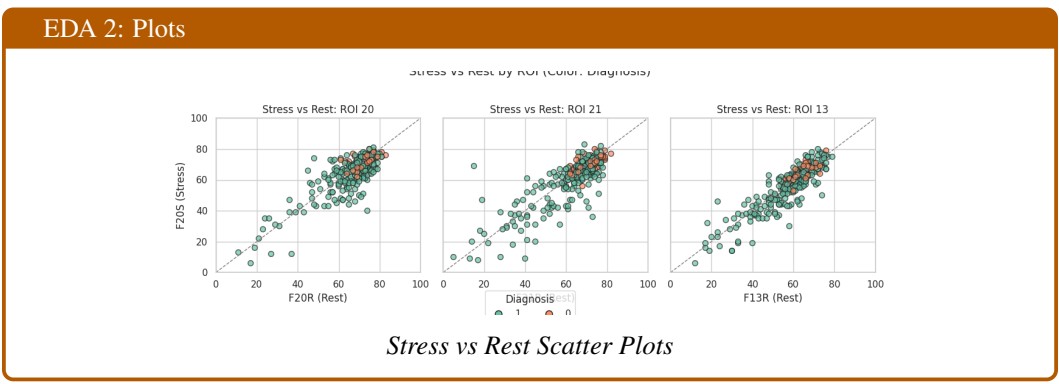

*Stress vs Rest Scatter Plots*

---

**EDA 2: Insights**

- Target balance (countplot in "Diagnosis Class Balance"): The dataset is imbalanced ( 210 of class "1" vs 55 of class "0"). This matters for feature choices that emphasize recall/precision tradeoffs and motivates variance-stabilizing transforms.

- Global ischemia separates classes (violin plots "Ischemic Burden Sum by Diagnosis" and "Worst Deficit by Diagnosis"): Both ischemic_burden_sum and worst_deficit are markedly higher and more rightskewed for class "1", with long heavy tails. The sqrt version reduces skew while preserving ordering.

- Top monotonic signals (Spearman barplot "Top Spearman Correlations with Diagnosis"):

  - Strong negatives: weak_triprod_top3 (0.47), F20S (0.41), MinShare_top3 (0.39), F21S (0.37), WorstStress_top3 (0.37), F3S (0.35), F13S (0.34). Interpretation: lower stress counts (especially in ROIs 20/21/13) and low weakest-link stress/share are associated with class "1".

  - Strong positives: burden_over_weakstress (+0.41), worst_deficit_over_WorstStress_top3 (+0.40), worst_deficit and its sqrt ( +0.36), ischemic_burden_sum and its sqrt ( +0.35), WorstRelDef_top4 (+0.34). Ratios that normalize deficits by weak stress are particularly discriminative.

---

**Feature Suggestions.**

**1. Global Stress–Rest Features**

- **Features:** `F1S–F22S`, `F1R–F22R`

- **Transformations:**

$$\text{global\_stress\_total} = \sum_{k=1}^{22} FkS$$

$$\text{global\_rest\_total} = \sum_{k=1}^{22} FkR$$

$$\text{global\_SR\_ratio} = \frac{\text{global\_stress\_total}}{\text{global\_rest\_total}}$$

- **Rationale:** Provides a global ischemic signature independent of ROI-level noise.

**2. Weakest-Link Ratio Features (ROIs 20, 21, 13)**

- Compute: `F20S/F20R`, `F21S/F21R`, `F13S/F13R`

- **Transformations:**

  - `MinSoverR_top3` = minimum of the three ratios
  - `MaxSoverR_top3` = maximum
  - `RatioRange_top3` = max min

- **Rationale:** Encodes focal ischemia (low minima), heterogeneity (range), and ROI-specific vulnerabilities.

**3. Global Burden Normalization**

- **Feature:** `burden_share_sqrt = sqrt_abs(ischemic_burden_sum) / global_stress_total`

- **Rationale:** Captures the global–local contrast consistent with heavy-tailed ischemic burden distributions.

**EDA Report**

**Code:**

```python
import pandas as pd
import numpy as np
import matplotlib.pyplot as plt
import seaborn as sns

sns.set_theme(style="whitegrid", context="notebook")

# Sanity checks
if 'df' not in globals():
    raise ValueError("DataFrame 'df' is required in the environment
        .")
if 'income' not in df.columns:
    raise ValueError("Target column 'income' is missing from 'df'."
        )

# Ensure target is numeric 0/1
if not np.issubdtype(df['income'].dtype, np.number):
    raise ValueError("Target 'income' must be encoded as 0/1
        numeric.")

# Baseline positive rate
baseline_rate = df['income'].mean()
print(f"Baseline >50K rate: {baseline_rate:.3f}")

# Helper: safe quantile binning with fallback
def safe_qcut(series, q=6, labels=None):
    """Quantile bin with duplicate handling; fallback to cut if
        needed."""
    s = pd.Series(series).astype(float)
    # Build quantile edges and drop duplicates
    quantiles = np.unique(np.nanpercentile(s, np.linspace(0, 100, q
        + 1)))
    # If too few unique edges, fallback to equal-width cut
    if len(quantiles) <= 2:
        eps = 1e-6
        return pd.cut(s, bins=q, duplicates='drop', include_lowest=
            True)
    try:
        return pd.qcut(s, q=min(q, len(quantiles) - 1),
                       duplicates='drop', labels=labels)
    except Exception:
        return pd.cut(s, bins=q, duplicates='drop', include_lowest=
            True)

# Helper: min/max cell summary for pivoted mean target
def summarize_pivot_rates(df_pivot, df_counts=None,
                          min_count=30, label=""):
    pivot_vals = df_pivot.copy()
    if df_counts is not None:
        mask = (df_counts >= min_count)
        pivot_vals = pivot_vals.where(mask)
    vmin = np.nanmin(pivot_vals.values)
    vmax = np.nanmax(pivot_vals.values)
    print(f"[{label}] cell >50K rate: min={vmin:.3f}, "
          f"max={vmax:.3f}, spread={vmax - vmin:.3f}")
```

**Output:**

```
Baseline >50K rate: 0.239
```

**EDA Report**

**Code:**

```python
# Age   Education interaction heatmap

if all(c in df.columns for c in ['age','education_num']):
    age_bins = safe_qcut(df['age'], q=6)
    edu_bins = safe_qcut(df['education_num'], q=6)

    # Range labels
    age_bins = age_bins.cat.rename_categories([f"{int(i.left)}   {
        int(i.right)}"
                                                for i in age_bins.
                                                    cat.categories])
    edu_bins = edu_bins.cat.rename_categories([f"{int(i.left)}   {
        int(i.right)}"
                                                for i in edu_bins.
                                                    cat.categories])

    df_tmp = df.assign(age_bin=age_bins, edu_bin=edu_bins)

    rate = df_tmp.pivot_table('income','edu_bin','age_bin','mean')
    cnt  = df_tmp.pivot_table('income','edu_bin','age_bin','size')

    fig, ax = plt.subplots(figsize=(9,5))
    sns.heatmap(rate, vmin=0, vmax=1, cmap="viridis",
                annot=True, fmt=".2f", cbar_kws={'label':'P(>50K)'
                    }, ax=ax)
    ax.set(title="Income rate: Education   Age",
           xlabel="Age bins", ylabel="Education_num bins")

    summarize_pivot_rates(rate, cnt, min_count=50, label="
        Age Education")
else:
    print("Required columns missing.")
```

**Output:**

```
[Age Education] cell >50K rate: min=0.007, max=0.649, spread=0.642
```

## A.7 EXTENDING PIFE TO DEEP LEARNING MODELS

Generalization across predictive model is crucial for flexibility of the pipeline. In this section, we will discuss the performance of MLP and TabPFN Hollmann et al. (2022). Using gpt-5 as both LLM and VLM, we have seen consistent improvements in MLP and TabPFN.

Table 10: Performance comparison across different neural predictive models. Values represent mean ± standard deviation of the metric score. We use F1-micro for classification and (1 - rae) for regression. For adult we have mentions N/A, TabPFN does not work for datasets larger than 10000 samples tasks. gpt-5 is used as both LLM and VLM.

| Dataset | MLP | | | TabPFN | | |
|---------|-----|-----|-----|--------|-----|-----|
| | Baseline | PIFE (Ours) | % Improvement | Baseline | PIFE (Ours) | Improvement |
| pima_indian[*] | $0.720 \pm 0.008$ | $\mathbf{0.731} \pm \mathbf{0.006}$ | 1.528 | $0.760 \pm 0.005$ | $\mathbf{0.768} \pm \mathbf{0.008}$ | 1.053 |
| fertility[*] | $0.840 \pm 0$ | $\mathbf{0.857} \pm \mathbf{0.025}$ | $2.024 \pm 0.000$ | $\mathbf{0.880}$ | $\mathbf{0.880}$ | 0 |
| housing_boston[†] | $0.678 \pm 0.002$ | $\mathbf{0.694} \pm \mathbf{0.006}$ | 2.36 | $0.736 \pm 0.002$ | $\mathbf{0.737} \pm \mathbf{0.005}$ | 0.136 |
| airfoil[†] | $0.728 \pm 0.003$ | $\mathbf{0.780} \pm \mathbf{0.003}$ | 7.143 | $\mathbf{0.886} \pm \mathbf{0.002}$ | $0.884 \pm 0.002$ | -0.226 |
| openml_586[†] | $0.582 \pm 0.006$ | $\mathbf{0.771} \pm \mathbf{0.024}$ | 32.474 | $0.855 \pm 0.002$ | $\mathbf{0.857} \pm \mathbf{0.001}$ | 0.234 |
| adult[*] | $0.851 \pm 0.002$ | $\mathbf{0.852} \pm \mathbf{0.001}$ | 0.118 | N/A | N/A | N/A |

### A.7.1 FEATURE IMPORTANCE IN NEURAL NETWORK MODELS

Neural network–based predictors do not provide intrinsic feature-importance scores. To integrate them into our framework, we approximate feature importance using model-specific post-hoc techniques.

**MLP Feature Importance.** For a Multilayer Perceptron (MLP), we compute feature importance from the first linear layer. Let the input dimension be $d$, and let the weight matrix of the first layer be

$$W^{(1)} \in \mathbb{R}^{h \times d},$$

where $h$ is the number of hidden units. The importance score for feature $j$ is defined as the mean absolute contribution across all hidden units:

$$\text{FI}_j^{\text{MLP}} = \frac{1}{h} \sum_{i=1}^{h} \big| W_{ij}^{(1)} \big|.$$

**TabPFN Feature Importance via Permutation.** For TabPFN, which is a black-box predictor, we estimate feature importance using permutation importance. Given a dataset $X \in \mathbb{R}^{n \times d}$ with labels $y$ and a predictive model $f$, let

$$\mathcal{S}(f, X, y)$$

denote the evaluation score (e.g., f1, 1-rae). For each feature $j$, we construct a perturbed dataset $X_{\text{perm}}^{(j)}$ by permuting only column $j$:

$$X_{\text{perm}}^{(j)} = \text{PermuteColumn}(X, j).$$

The feature importance is quantified as the drop in performance due to permutation:

$$\text{FI}_j^{\text{TabPFN}} = \mathcal{S}(f, X, y) - \mathcal{S}(f, X_{\text{perm}}^{(j)}, y).$$

Higher values indicate stronger contribution of feature $j$ to predictive performance.

### A.8 REVERSE POLISH NOTATION FOR FEATURE REPRESENTATION

We adopt Reverse Polish Notation (RPN) from Zou et al. (2025) as a representation scheme for the features generated in PIFE. In RPN, operators follow their operands, eliminating the need for parentheses and reducing ambiguity in expression evaluation. This structure allows for a compact and unambiguous encoding of complex feature transformations, which is particularly useful when features are generated programmatically or by language models.

Using RPN provides several advantages. First, it enables straightforward reconstruction of the original feature expression, as the sequence of operands and operators directly encodes the computational order. Second, RPN facilitates efficient storage and manipulation of features, since it can be easily parsed into computational graphs or evaluated using stack-based execution.

| Comparison | t-statistic | p-value | Interpretation |
|---|---|---|---|
| CAAFE vs OCTREE | 3.6698 | 0.0009 | CAAFE performs significantly better than OCTREE |
| CAAFE vs PIFE | −2.4065 | 0.0216 | PIFE performs significantly better than CAAFE |
| OCTREE vs PIFE | −5.0073 | $1.10 \times 10^{-5}$ | PIFE performs significantly better than OCTREE |

Table 11: Welch's t-test results comparing CAAFE, OCTREE, and PIFE across datasets to assess statistical significance.

### A.9 STATISTICAL SIGNIFICANCE ACROSS BASELINE METHODS

**Statistical Significance Analysis.** We assess whether the performance improvements of PIFE over the baseline AutoFE method are statistically meaningful by conducting a Welch's t-test at a significance level of $\alpha = 0.05$. As reported in Table 11, PIFE achieves statistically significant gains with notable improvements on more complex or high-dimensional tasks such as *airfoil*, *megawatt_1*, *messidor_features*, and several OpenML benchmarks. These are domains where multi-step feature reasoning and interaction-driven transformations are particularly beneficial, and the iterative EDA-guided process of PIFE provides measurable advantages.

For the remaining datasets, the performance differences are not statistically significant; however, PIFE matches or slightly exceeds the baseline across all cases, indicating method stability and the absence of regressions. Importantly, PIFE does not exhibit statistically significant degradation on any dataset. Overall, the significance analysis confirms that PIFE delivers robust improvements and is especially effective in settings where higher-order feature interactions play a critical role.

### A.10 HYPERPARAMETERS

#### A.10.1 PREDICTIVE MODELS

As shown in Table 12, we consider both regression and classification models with standard hyperparameters for baseline evaluation and testing AutoFE methods.

Table 12: Regression and Classification Models with Parameters

| Task | Model | Parameters |
|---|---|---|
| Regression | Linear Regression | – |
| Regression | Random Forest Regressor | n_estimators = 10, random_state = 0 |
| Regression | XGBoost Regressor | n_estimators = 10, random_state = 0 |
| Classification | Logistic Regression | solver = saga, class_weight = balanced, tol = 0.0005, C = 0.5, max_iter = 10000, penalty = l2 |
| Classification | Random Forest Classifier | n_estimators = 10, random_state = 0 |
| Classification | XGBoost Classifier | n_estimators = 10, random_state = 0 |

#### A.10.2 PARAMETERS FOR DEEP LEARNING METHODS

We list down hyperparameters used for MLP and HyperFast in Table 13 and Table 14.

Table 13: Parameters for MLP

| Parameter | Value |
|---|---|
| Number of layers | 3 |
| Layer size | 256 |
| Dropout | 0.2 |
| Learning rate | $1 \times 10^{-3}$ |
| Batch size | 256 |
| Epochs | 100 |
| Optimiser | Adam |
| Patience | 40 |

Table 14: Parameters for HyperFast

| Parameter | Value |
|---|---|
| Number of ensembles (N) | 16 |
| Batch size | 2048 |
| NN bias | False |
| Stratified sampling | False |
| Optimization strategy | None |
| Optimize steps | 64 |
| Random seed | 3 |

### A.10.3 FEATURE SELECTION METHODS

Details of parameters used for feature selection experiments (results in Table 3) are listed in Table 15 and Table 16.

Table 15: Parameters for Bayesian CMI-based Feature Selection

| Parameter | Values |
|---|---|
| alpha | 0.5 |
| trials | 50 |
| distance | gower |
| min_num_feat_selected | 0 |
| scaling_criteria | min_max |
| sample_df | True |
| k | 10 |
| denomination | 2 |
| num_select_features | 3 |

Table 16: Parameters for Genetic Algorithm-based Feature Selection

| Parameter | Values |
|---|---|
| elitism | 5 |
| generations | 30 |
| population_size | 30 |
| crossover_prob | 0.8 |
| mutation_prob | 0.05 |

### A.10.4 AUTOFE METHODS

Table 17 summarizes the key parameters and their values for the AutoFE methods evaluated in this study, including PIFE, OCTREE, CAAFE, OPENFE, AUTOFEAT, and DFS. These values were chosen based on prior literature and preliminary experiments to ensure fair and comparable evaluation across methods.

### A.11 BASELINE SELECTION

While learning-based approaches, which aim to learn transformation policies directly from data, represent an important category, we do not include them in our current evaluations due to the high implementation complexity and substantial computational cost involved in training and adapting these models. This selection allows us to compare how different strategies perform in practice and to analyze their respective strengths, limitations, and implications for the future of feature engineering.

The original CAAFE framework was primarily designed for classification tasks and evaluated only with accuracy as the performance metric. In our adaptation, we extend CAAFE to also support

Table 17: Parameters and Values for AutoFE Variants

| Method | Parameter | Value |
|--------|-----------|-------|
| PIFE | total_steps | 10 |
| | eda_steps | 3 |
| | max_plots | 3 |
| | max_insights_per_plot | 1 |
| | n_features | 5 |
| | timeout | 86400 |
| OCTree | total_steps | 5 |
| | rule_steps | 10 |
| | n_features | 1 |
| | timeout | 86400 |
| CAAFE | total_steps | 10 |
| | n_features | 1 |
| | n_repeats | 1 |
| | timeout | 86400 |
| OpenFE | min_candidate_features | 2000 |
| | feature_boosting | False |
| | n_repeats | 1 |
| | timeout | 86400 |
| Autofeat | feateng_steps | 2 |
| | timeout | 86400 |
| DFS | max_depth | 2 |
| | transformations | transform_primitives |
| | timeout | 86400 |

regression tasks, introduce additional evaluation metrics beyond accuracy for a fairer comparison, and enrich the set of operators available for feature construction.

OCTree, on the other hand, required more substantial modifications. The original implementation was tightly coupled with specific LLM APIs and lacked iterative refinement. We restructured its pipeline to generalize API usage, extended the feature generation loop to be iterative, and incorporated support for regression tasks along with additional evaluation metrics. These modifications make OCTree more robust and applicable across a broader range of tabular learning scenarios.

### A.12 DISCUSSION ON FEATURE SELECTION METHODOLOGIES

Feature selection in our framework can be applied at two stages: (1) immediately after the feature engineering step, or (2) after the full PiFE run. In our experiments, Feature Importance and Bayesian Conditional Mutual Information (CMI) based selection were applied after the feature engineering stage but prior to model validation, whereas a Genetic Algorithm based selection was performed after the complete pipeline execution.

Feature selection plays a crucial role in enhancing both model interpretability and generalization. While an individual engineered feature may appear weak in isolation, its combination with other features can capture complex interactions and yield a much stronger predictive signal. Without an appropriate selection mechanism, such subtle but useful interactions may be overlooked or drowned out by a large number of irrelevant or redundant features. By systematically ranking and filtering features, our selection strategies help retain those that contribute jointly to predictive power, thereby improving efficiency, reducing overfitting, and uncovering more meaningful feature representations.

### A.12.1 FEATURE IMPORTANCE BASED SELECTION

After features are generated in an iteration , we perform validation on the dataset with new features and compute feature importance scores. If the validation score of the current run is greater than

that of the previous run, we retain all the features. Otherwise, we apply a filtering criterion: only those features with importance greater than 1 / (number_of_features) are selected. This threshold is motivated by the expectation that a retained feature should contribute at least more than the average share of importance across all features.

### A.12.2 CMI-BASED BAYESIAN OPTIMIZATION FOR FEATURE GROUP SELECTION

**Bayesian Optimization (BO).** BO is a sequential strategy for optimizing expensive black-box functions. A surrogate model (e.g., Gaussian Process) provides mean $\mu(x)$ and uncertainty $\sigma(x)$, guiding the selection of new points via an acquisition function $a(x)$ (e.g., EI, UCB):

$$x_{t+1} = \arg\max_{x \in \mathcal{X}} a(x \mid \mu(x), \sigma(x)).$$

In feature engineering, we need to add a feature subset to the the existing feature set. BO treats this feature subset as $X$ and the CMI as $f(X)$(objective function), enabling efficient exploration of feature combinations.

$$f(X) = I(X; Y \mid Z),$$

where $X$ is the feature subset, $Y$ is the target, and $Z$ is the feature set.

**Conditional Mutual Information (CMI).** CMI in feature engineering quantifies the unique contribution of a feature subset $X$ to predicting $Y$ given the feature set $Z$:

$$I(X; Y \mid Z) = \iint \int p(x, y, z) \log \frac{p(x, y \mid z)}{p(x \mid z)\, p(y \mid z)} \, dx \, dy \, dz.$$

We use a slightly modified version of the mixed-type $k$-NN estimator from Mesner & Shalizi (2020), which is robust to discrete and continuous variables. We set $K = \max(3, \min(20, \sqrt{n}))$, which helps mitigatethe high-dimensionality issue in CMI calculation.

### A.12.3 GENETIC ALGORITHM

Genetic Algorithms (GAs) are population-based metaheuristic optimization methods. A GA maintains a population of candidate solutions (chromosomes), where each chromosome encodes a subset of features (typically as a binary string, with 1 indicating selection of a feature and 0 otherwise). The algorithm evolves this population through the iterative application of genetic operators:

- **Selection:** Chromosomes are chosen based on their fitness, which in our case is the predictive performance (e.g., validation accuracy or $F_1$ score)
- **Crossover:** Pairs of chromosomes exchange parts of their feature subsets, enabling exploration of new feature combinations.
- **Mutation:** Random flips of feature bits introduce diversity and help escape local optima.

The fitness of a chromosome $c$ can be expressed as

$$\text{Fitness}(c) = \text{Score}\big(f(X_c), Y\big),$$

where $X_c$ denotes the features selected by chromosome $c$, $Y$ is the target, and $f(\cdot)$ is the downstream predictive model.

Over successive generations, the GA converges toward feature subsets that maximize predictive performance. While computationally more expensive than other methods like CMI-BO and MFI, GAs often identify subsets of features with strong predictive power, making them effective when interactions between features play an important role.

### A.13 PROMPTS

Listing 7: EDA Analysis Code Generation

```
You are an EDA agent operating in a Kaggle Grandmaster-style automated
    feature engineering pipeline. This is iteration {current_iteration}
    of a multi-stage EDA loop. Your role is to produce competition-grade
    exploratory data analysis code that progressively builds upon the
    analyses performed in previous iterations.

Remember: this is a strategic, hypothesis-driven EDA process - think
    like a top Kaggle competitor uncovering hidden signal iteratively
    across multiple passes.
```

```
You are provided with:
- Dataset description
- Summary of preprocessing steps taken
- Previous EDA code history and respective observations if any.

Dataset Description:
{dataset_description}

Pre-processing Steps:
{preprocessing_summary}

Focus Areas: 0/1/2

% focus_strategies = {
%     0: {
%          "primary": "distribution_analysis",
%          "secondary": "correlation_analysis",
%          "description": "Initial exploration: distributions and basic
    correlations of top features"
%     },
%     1: {
%          "primary": "interaction_analysis",
%          "secondary": "non_linear_patterns",
%          "description": "Interaction exploration: feature pairs and
    non-linear relationships"
%     },
%     2: {
%          "primary": "temporal_categorical",
%          "secondary": "outlier_analysis",
%          "description": "Advanced patterns: temporal trends and
    categorical encodings"
% }
}

Memory (previous code and observations):
{memory}

---

### Analysis Constraints
STRICT LIMITS:
- Maximum {max_plots} plots total
- Do NOT explicitly suggest feature transformations, binning, encoding,
    or normalization.
- Focus on uncovering patterns, trends, correlations, and anomalies in
    the data.
- Avoid bias towards only high-importance features  include a mix of
    numerical, categorical, and temporal features.
- For large datasets (>5000 rows), sample strategically before complex
    plots.

---

### Response Format
Your response should strictly follow the following Code Structure:

Before the code block include a short **Implementation Rationale** of 24
    sentences that explains:
- Why you chose the specific analyses / plots (what hypothesis you are
    testing),
- What you expect the output to reveal (the type of insight sought),
- One-line failure mode / limitation: Think Harder (e.g., 'may fail on
    heavy-tailed column; will downsample if >5000 rows').
```

```
- Divide the code into code cells using `# %%` to demarcate different
    sections in the code.
- Code should be wrapped within ```python ... ``` quotes. Do not write
    code at any other place than this.
- Do not write the context of the code block in the same line as `# %%`.
    Write it in a new line with enumeration, where enumeration should be
    in comments.
- Each section should focus on a different type of analysis aimed at
    revealing meaningful patterns.
- Ensure diversity by analyzing features across different types and
    varying levels of correlation with the target.
- All plots must have proper titles, axis labels, and legends where
    applicable.
- Produce plots that are clear and informative, suitable for
    presentation.
- If plots become too crowded or contain too many elements to be
    readable, split them into multiple smaller plots.
- Each plot should be individually assigned to a unique and
    human-readable variable name.
- Do NOT use `plt.show()` or save figures to files  just generate them.
- For expensive plots (swarm, violin, scatter, kde, displot, etc.), if
    dataset size >5000 rows, downsample to 5000 using stratified
    sampling (if categorical column available), else use appropriate
    sampling.

---

### Mandatory Imports
Each EDA code block must begin with:
```python
import pandas as pd
import numpy as np
import matplotlib.pyplot as plt
import seaborn as sns

---

### General Instructions

- Assume the dataset with variable name df is already present.
- Always include all required imports in your code. Do NOT assume
    imports persist across steps.
- Preserve existing variable assignments and do not overwrite previously
    assigned variables.
- Maintain continuity with past EDA steps by avoiding duplicate analyses
    and expanding on previously explored patterns.
- Incorporate findings from earlier iterations to guide where to dig
    deeper.
- Ensure coverage across diverse feature types (numerical, categorical,
    temporal).
- Generate new, non-redundant insights that add incremental value.
- All generated code should be clean, modular, and ready for execution
    without edits.
- Use # %% to clearly separate analysis sections.
- Include explanatory comments in the format: # INSIGHT: <purpose of
    this analysis>, but do NOT explicitly suggest feature
    transformations.
```

Listing 8: EDA Analysis Insight Generation

```
You are a feature engineering specialist analyzing outputs from an EDA
    process.
You are provided with:
```

```
- The code used to generate the EDA
- The plots generated from that code
- The textual/statistical outputs produced

Your job is to extract meaningful, high-value insights from these
    materials and then propose specific, well reasoned feature
    transformations inspired by these insights. Insights must be
    grounded in visual and statistical evidence, not speculation.

Think like a Kaggle Grandmaster preparing features for a top-tier
    competition.

# Task Description
{dataset_description}

# EDA Code Context
{eda_code}

# Available Operators
{operators_description}

# Instructions

## Analysis Guidelines When forming INSIGHTS:
- Highlight relationships, anomalies, patterns, or distributions that
    stand out in the data.
- Capture interactions, trends, and category-level differences.
- Refer to the specific visualization or statistical summary they come
    from.
- Generate maximum {max_insights_per_plot} insights per plot.

## When forming FEATURE_TRANSFORMATIONS:
- Map each transformation to a corresponding insight.
- Use ONLY the Available Operators listed above for suggesting feature
    transformations.
- You can suggest dropping features if they are redundant, highly
    correlated, or unhelpful for the target variable.
- Provide reasoning linked to potential model performance improvements.
- Keep recommendations actionable, clear, and technically precise.
- Strictly NEVER include the target column in any feature engineering,
    transformations, encodings, interaction terms, binning, scaling, or
    statistical computations.

# Response format

Your output MUST be structured in exactly two sections using the
    following XML-style tags:

<INSIGHTS>
List clear, evidence-backed observations from the EDA results and plots.
Avoid feature suggestions here  keep this purely as descriptive,
    analytical findings.
Each insight should explicitly reference the plot or analysis it came
    from.
</INSIGHTS>

<FEATURE_TRANSFORMATIONS>
For each transformation:
- Specify the exact feature(s) involved
- Describe the suggested transformation or engineering step using
    Available Operators
- Provide a short reasoning for why it is beneficial based on the
    insights above
- Include 35 high-priority transformations that would add the most value
```

```
- You can suggest dropping features if they are redundant, highly
    correlated, or unhelpful for the target variable.
</FEATURE_TRANSFORMATIONS>

AVOID: generic or obvious patterns, restating axis labels, or vague
    statements.
FOCUS: insights that directly inform strong, competition-grade feature
    engineering.
```

Listing 9: Feature Engineering Code Generation

```
You are a feature-engineering agent. Your goal is to generate new
    machine-learning-ready features and return executable Python code
    that creates them.You will be provided with the Dataset Description,
    the Pre-processing Steps, the Feature Transformation Guidelines,
    list of allowed operators for feature engineering, the Feature
    importance scores and the Rejected Features.

# Task description:
{dataset_description}

# Pre-processing Steps:
{preprocessing_step_summary(preprocess,target_encoder)}

# Feature Transformation Guidelines:
{guidelines}

# Available Operators:
{operators_description}

# Feature importance scores:
{[f"{k}: {v:.5f}" for k, v in feature_importance_scores.items()] if
    feature_importance_scores is not None else []}

# Rejected Features:
{rejected_features}

# Instructions:

## Feature Engineering Instructions:
1) Generate exactly {n_features} new machine-learning-ready features.
2) You should think about the reasons for the rejected features and try
    to incorporate that learning while creating new featurs
3) Feature Transformation Guidelines contains ideas based on exploratory
    data analysis conducted earlier. You should create new features that
    are based on the ideas in the Feature Transformation Guidelines.
4) You should use the list of  Available Operators, for feature
    engineering to create new features.

##  Code Instructions:
- Do not wrap the entire code inside a function or class.
- Assume the environment is similar to a Jupyter Notebook, so you may
    use # %% to separate code blocks.You may define small utility/helper
    functions if needed, but make sure they are invoked within the same
    code block.
- The final output should be an executable code block, not a function or
    class definition.
- Ensure that code is enclosed within python code literal as follows. Do
    not write code anywhere else.
```python
[code goes here]
```

# Target Leakage Prevention Rules:
```

```
STRICT RULES TO AVOID TARGET LEAKAGE:

- NEVER create features that directly transform or encode the target
    column itself (e.g., log(target), residuals vs. target,
    deviation-from-target, ranks of target within bins, z-scores,
    percentiles, etc.).
- The target column may only be used to compute group-level aggregate
    statistics (mean, median, std, count) at the group level.
    - Example: mean(target) by year, median(target) by region.
    - These must be computed on df_train only, stored as a mapping, and
    applied to df_test with a fallback.
- Forbidden feature patterns include:
    - Any function that directly transforms target values row-by-row
    (log, rank, residual).
    - Any feature whose definition requires subtracting or dividing a
    rows own target from an aggregate.
    - Any within-bin or within-group ranking of target values.
    - Always check that the engineered feature can be computed in
    exactly the same way for both df_train and df_test without needing
    df_test[TARGET_COL].
    - If a proposed feature would violate these rules, DO NOT generate
    it  instead, list it in
    SAFETY_REPORT['features_dropped_due_to_no_test_support'].

# Feature Inclusion & Target-Use Rules (MANDATORY, concise):
CRITICAL: The runtime convention is that the target column WILL be
    present in df_train as 'target', and WILL NOT be present in df_test.
    Follow these rules without exception:

- INPUT ASSUMPTION
    - df_train contains the target column named 'target'.
    - df_test, if provided, MUST NOT contain the target column. Agents
    must treat df_test as unlabeled.
- FEATURE INCLUSION: Every suggested feature must be constructible on
    BOTH df_train and df_test. If the feature cannot be created for
    df_test without reading the target (TARGET_COL) or other unavailable
    test-only data, DO NOT create that feature for df_train  drop it
    entirely. Never produce train-only features.

- TARGET USAGE (TRAIN-ONLY STATISTICS): You may compute aggregate
    statistics using df_train[TARGET_COL] only to build train-derived
    mapping objects (e.g., group means/counts/medians). All such
    computations MUST:
    - be computed only on df_train,
- APPLYING MAPPINGS TO TEST: For every train-derived mapping, provide
    explicit code that applies the mapping to df_test inside if df_test
    is not None: using map/merge and .fillna(<fallback>).
    - Example pattern (must be used):
    ```python
    mapping = df_train.groupby('X')[TARGET_COL].median().to_dict()
    df_train['f'] = df_train['X'].map(mapping).fillna(<fallback>)
    if df_test is not None:
        df_test['f'] = df_test['X'].map(mapping).fillna(<fallback>)
    _train_mappings['mapping_name'] = mapping
    ```
- INTERMEDIATE / TRAIN-ONLY COLUMNS: If you create intermediate columns
    on df_train solely to compute mapping/statistics or to use them to
    create suggested features, remove them from df_train before
    finishing the code (so columns remain symmetric). Do NOT leave
    intermediate columns that cannot be created on df_test.
- COLUMN SYMMETRY CHECK: At the end of your code, ensure df_train and
    df_test have the same columns (except for TARGET_COL in df_train).
    Add this assertion:
    ```python
```

```
        # Ensure column symmetry between train and test sets
        if df_test is not None:
            train_cols = set(df_train.columns) - {TARGET_COL}
            test_cols = set(df_test.columns)
            assert train_cols == test_cols, f"Column mismatch: train has
    {train_cols - test_cols} extra, test has {test_cols - train_cols}
    extra"
        ```
- NON-IMPLEMENTABLE FEATURES: If any transform (e.g., direct arithmetic
    with TARGET_COL in df_test, or features requiring target values at
    test-time) cannot be implemented safely on df_test, explicitly
    exclude that feature and list it under
    SAFETY_REPORT['features_dropped_due_to_no_test_support'].
- FORBIDDEN: Under no circumstance should the generated code access
    TARGET_COL within df_test or assume its existence there. Do not
    create features that would require test-time predictions or labels.
- RANDOMNESS: Use deterministic randomness via RANDOM_STATE for any
    sampling/splitting operations on df_train; do not sample df_test.

# Templates (must be used for any transform that depends on target/train
    statistics):
Provide transformations using these exact patterns when the operation
    depends on train statistics.

A) GroupByThenMean (safe pattern)
```python
# compute mapping on train ONLY
edu_mean_by_occ =
    df_train.groupby('occupation')['education-num'].mean().to_dict()
# apply to train
df_train['QualificationSurplus'] = df_train['education-num'] -
    df_train['occupation'].map(edu_mean_by_occ)
# apply to test (no target used). fallback to 0 for unseen occupations
if df_test is not None:
    df_test['QualificationSurplus'] = (
        df_test['education-num'] -
    df_test['occupation'].map(edu_mean_by_occ)
    ).fillna(0)
```

B) Target-like encoding (train-derived, safe pattern)
```python
# compute target-encoding stats on train ONLY
enc_by_cat = df_train.groupby('cat_col')[TARGET_COL].agg(
                ['mean','count']
            ).to_dict(orient='index')
# convert to mapping (use mean, with global fallback)
global_mean = df_train[TARGET_COL].mean()
cat_mean_map = {k: v['mean'] for k, v in enc_by_cat.items()}
# apply
df_train['cat_col_te'] =
    df_train['cat_col'].map(cat_mean_map).fillna(global_mean)
if df_test is not None:
    df_test['cat_col_te'] =
    df_test['cat_col'].map(cat_mean_map).fillna(global_mean)
```

C) Stratified sampling for train-only operations (must not touch df_test)

```python
from sklearn.model_selection import StratifiedKFold
skf = StratifiedKFold(n_splits=5, shuffle=True,
    random_state=RANDOM_STATE)
for train_idx, holdout_idx in skf.split(df_train, df_train[TARGET_COL]):
```

```
    # operate only on df_train.iloc[train_idx], use holdout for internal
    validation
    pass
```

D) ALWAYS include mapping objects in code and show how they'll be
    persisted/serialized if needed.

# Templates for Intermediate Features:

```python
Interaction: lymphatics  early_uptake (concatenated string, factorized)
df_train['lymphatics_earlyuptake'] = df_train['lymphatics'].astype(str)
    + '_' + df_train['early_uptake'].astype(str)
# Factorize (shared mapping for train, then reapply to test)
all_cats = pd.concat([df_train['lymphatics_earlyuptake'],
                     (df_test['lymphatics'].astype(str) + '_' +
    df_test['early_uptake'].astype(str)) if ('df_test' in locals() and
    df_test is not None) else pd.Series([],dtype=str)]
lympt_early_map, lympt_early_uniques = pd.factorize(all_cats, sort=True)
train_codes = lympt_early_map[:len(df_train)]
df_train['lymphatics_earlyuptake_code'] = train_codes
# Remove lymphatics_earlyuptake as its not the suggested feature and it
    can not be created in df_test
df_train.drop(columns=['lymphatics_earlyuptake'], inplace=True)
if 'df_test' in locals() and df_test is not None:
    test_codes = lympt_early_map[len(df_train):]
    df_test['lymphatics_earlyuptake_code'] = test_codes
_train_mappings['lymphatics_earlyuptake_factorization'] =
    dict(zip(lympt_early_uniques, range(len(lympt_early_uniques))))
```

– IMPORTANT: Intermediate Feature Cleanup
At the end of your code, ensure you remove ALL intermediate features
    that were created solely for computation purposes:

```
# Clean up intermediate features
intermediate_features = ['temp_feature1', 'temp_feature2',
    'mapping_temp']
for feature in intermediate_features:
    if feature in df_train.columns:
        df_train.drop(columns=[feature], inplace=True)
        if df_test is not None and feature in df_test.columns:
            df_test.drop(columns=[feature], inplace=True)
```

# Response Format for Python Code:

– Python code for n feature transformations

```python
[feature engineering code]
```

– RPN Format:

Reverse Polish Notation and Description of n feature transformations
Provide in this format:

FeatureName: <new_feature_name>
RPN : feature1 feature2 +
Description : Sum of feature1 and feature2

FeatureName: <new_feature_name>

```
RPN : feature3 feature3 feature4 GroupByThenMean -
Description : Difference between feature3 and mean of feature3 grouped
    by feature4

# Instructions for RPN Notation:

- Use the format Dropped_<FeatureName> for features that are dropped.
- Do not use square brackets in the FeatureName.
- Separate tokens in the RPN string with spaces.
- Examples of correct RPN:
    - feature1 feature2 +
    - feature1 drop (for dropping a feature)
    - feature1 feature1 feature2 GroupByThenMean -

- Avoid invalid RPN such as feature1 feature2 GroupByThenMean -
    subtraction - requires two operands.

- Example with GroupByThenMean:

    '''python
    edu_mean_by_occ =
    df_train.groupby('occupation')['education_num'].mean()
    df_train['QualificationSurplus'] = df_train['education-num'] -
    df_train['occupation'].map(edu_mean_by_occ)
    python
    Copy code
    edu_mean_by_occ =
    df_train.groupby('occupation')['education_num'].mean()
    df_train['QualificationSurplus'] = df_train['education-num'] -
    df_train['occupation'].map(edu_mean_by_occ)
    if df_test is not None:
        df_test['QualificationSurplus'] = (
            df_test['education-num'] -
    df_test['occupation'].map(edu_mean_by_occ)
        ).fillna(0)
    '''

    RPN: education-num education_num occupation GroupByThenMean -
    This RPN correctly represents the operation.
    Incorrect RPN: education_num occupation GroupByThenMean - (only 1
    operand before subtraction)

- Drop Operation Examples:
    - To drop a feature: RPN: feature_name drop
    - To drop multiple features: RPN: feature1 drop feature2 drop
    - Always document dropped features in your response with the
    Dropped_<FeatureName> format
```

A.14   EXAMPLE PIFE FEATURES FROM EXPERIMENT RUNS

---

**Higher Order Feature**

**Competition** : spectf
**Name of the feature**: Sum_x_Hotspot
**RPN**: F1S F2S + F3S + F4S + F5S + F6S + F7S + F8S + F9S + F10S + F11S + F12S + F13S + F14S + F15S + F16S + F17S + F18S + F19S + F20S + F21S + F22S + F20S F21S max F22S F13S max max *
**Order**: 21
**EDA Reasoning**:   Interaction of global stress and regional hotspot (maps to the "global–regional synergy" insight)
**Features**: F1S,...,F22S, max_stress_13_20_21_22
Transformation:
1) StressSum_all = F1S + F2S + ... + F22S (chain the "+" operator across all stress ROIs)
2) Sum_x_Hotspot = StressSum_all  max_stress_13_20_21_22
**Reasoning**: The Q1 quadrant (high-high) showed a 0.43 abnormal rate vs 0.20 elsewhere. The multiplicative term encodes this synergy explicitly and is often more predictive than either marginal.

---

Figure 7: Higher Order Feature Generation

---

**Higher Predictive Power**

**Competition**: poker_hand
**Name of the feature**: rank_pair_sum
**RPN**: C1 C1 / C1 C2 - abs + reciprocal round C1 C1 / C1 C3 - abs + reciprocal round + C1 C1 / C1 C4 - abs + reciprocal round + C1 C1 / C1 C5 - abs + reciprocal round + C2 C1 - abs C1 C1 / + reciprocal round + C2 C3 - abs C1 C1 / + reciprocal round + C2 C4 - abs C1 C1 / + reciprocal round + C2 C5 - abs C1 C1 / + reciprocal round + C3 C4 - abs C1 C1 / + reciprocal round + C3 C5 - abs C1 C1 / + reciprocal round + C4 C5 - abs C1 C1 / + reciprocal round +
**Feature Importance**: 0.35005184128253863
Increase in score after adding feature: 0.306
**EDA Reasoning** :
**Feature(s)**: C1–C5
**Transformation**: Build pairwise "same-rank" indicators and aggregate. 1) ONES = C1 / C1 2) For each unordered pair (i, j) among 1..5: diff_ij = abs(Ci  Cj) diff1_ij = diff_ij + ONES eq_ij = round(reciprocal(diff1_ij))  equals 1 if Ci=Cj, else 0 3) rank_pair_sum = sum(eq_ij over the 10 pairs) using + 4) For each i: eq_i = sum(eq_ij over j  i) using + max_same_rank = max(eq_1, eq_2, eq_3, eq_4, eq_5) using max
**Reasoning**: From the correlation heatmap and uniform marginals, single ranks are uninformative; equality patterns drive CLASS. rank_pair_sum differentiates high-card/straight/flush (0), one pair (1), two pair (2), three-kind (3), full house (4), four-kind (6). max_same_rank (values 0–3) is a strong proxy for the largest multiplicity (pair/three/four). These directly target rare classes (3,6,7) and improve separability under heavy imbalance.

---

Figure 8: Higher Predictive Power

---

**Weak features combined to get strong feature**

**Competition**: messidor_features
**Name of the feature**: MA_RANGE
**RPN**: ma1 ma2 max ma3 max ma4 max ma5 max ma6 max ma1 ma2 min ma3 min ma4 min ma5 min ma6 min -
**Feature Importances**:
```
MA_RANGE:  0.09319
ma1:  0.04985
ma2:  0.04468
ma3:  0.04219
ma4:  0.03743
ma5:  0.04279
ma6:  0.03985
```

**EDA Reasoning**: MA_RANGE: max(ma1, max(ma2, max(ma3, max(ma4, max(ma5, ma6))))) min(ma1, min(ma2, min(ma3, min(ma4, min(ma5, ma6)))))
**Reasoning**: Despite high within-cluster correlations, thresholds are not identical. The box-plots show distributional spread growing with DR. The range across thresholds measures stability/sensitivity of detections to $\alpha$; noisy/non-DR images may show different spreads than true DR.

---

Figure 9: Effective Feature Combination

### A.15 COST AND TIME

Table 18 summarizes the average monetary cost of running PIFE using GPT-4.1 and GPT-5. The reported values include both the Exploratory Data Analysis (EDA) step and the feature engineering (FE) stage. When using GPT-4.1, the complete pipeline costs approximately \$1.21 per dataset, with \$0.94 spent on EDA and \$0.27 spent on FE. In contrast, the cost increases to approximately \$4.95 per dataset when using GPT-5, with \$4.15 attributed to EDA and \$0.79 to FE. Higher cost is expected because GPT-5 is a reasoning-oriented model and typically produces longer and more detailed analytical outputs, resulting in increased token usage.

Table 18: PIFE cost for gpt-4.1 and gpt-5

| LLM | EDA cost per dataset | FE cost per dataset | Cost per dataset |
|-----|---------------------|--------------------|-----------------|
| gpt-4.1 | \$ 0.93894 | \$ 0.26953 | \$ 1.2085 |
| gpt-5 | \$ 4.1549 | \$0.7905 | \$ 4.9454 |

Table 19 reports the average time required for EDA and feature engineering per dataset when using GPT-4.1 and GPT-5. Although GPT-5 incurs a significantly higher runtime (approximately 9,000 seconds per dataset), this difference is primarily attributable to its stronger reasoning capabilities, which lead to more detailed analyses and substantially longer generated outputs. In contrast, GPT-4.1 completes the entire pipeline in roughly 2,000 seconds per dataset, producing shorter and more concise reasoning chains.

Table 19: PIFE time(in s) for gpt-4.1 and gpt-5

| LLM | EDA time per dataset | FE time per dataset | Time per dataset |
|-----|---------------------|--------------------|-----------------|
| gpt-4.1 | 1438.71 | 378.99 | 2031.41 |
| gpt-5 | 7380.7 | 1307.16 | 8995.19 |

### A.16   LLM USAGE

Apart from our proposed framework, PiFE, we leveraged LLMs to assist in refining the writing of this research paper. The models were used solely for language polishing, grammar corrections, and clarity improvements, without influencing the scientific content, experimental design, results, or conclusions.

### A.17   BROADER IMPACT STATEMENT

**Utility and Real-World Relevance.** The AutoFE method can autonomously generate, evaluate, and select meaningful features from raw data, potentially enabling more robust and interpretable predictive models in real-world applications. By leveraging EDA-driven insights, it can provide data scientists with explainable and interpretable features, enhancing model transparency and decision-making. With appropriate statistical safeguards, validation checks, and privacy-preserving measures, it can help mitigate the risk of spurious or misleading features and support the responsible deployment of LLM-assisted feature engineering systems.

**Risks and Biases.** LLM-assisted feature generation can be influenced by biases present in the training data, including memorization of datasets or solutions from prior competitions, which may lead to overfitting or inflated performance on familiar tasks. To mitigate this, incorporating recent Kaggle competitions and unseen datasets during evaluation can help assess generalization and reduce reliance on memorized patterns. Selecting diverse datasets from multiple sources is critical to capture varied real-world scenarios, minimize systemic bias, and ensure broadly applicable and fair features. Additionally, running experiments across multiple random seeds can provide a more robust assessment of LLM-based frameworks, helping to quantify variability and improve reliability in feature generation.

