# OpenReview forum: "PIFE: Progressive Insight driven Feature Engineering via Multimodal Reasoning"
_ICLR.cc/2026/Conference — Submitted to ICLR 2026_

### Official Review · Reviewer_96TH · 2025-10-30

**Soundness:** 2
**Presentation:** 4
**Contribution:** 2
**Rating:** 4
**Confidence:** 2

**Summary:**

The paper proposes an automated feature engineering framework that involves iteratively performing exploratory feature analysis and feature generation, guided by large language models. The method is evaluated in a number of tabular datasets, and ablation studies are performed.

**Strengths:**

- The approach is interesting, the idea that LLMs can guide EDA and extract features that are more effective at downstream tasks is explained well.
- The paper is written well and easy to follow.

**Weaknesses:**

- The use of a VLM is questionable to me, I think the method tries to mimic how a human would go about the process of exploratory data analysis and feature engineering. However, if the goal is to improve performance on downstream tasks, (which most of the evaluation focuses on) the benefit of using large models is their ability to understand complex relationships within the data, where generating plots would be unnecessary.
- Table 2 shows that in the experiments performed, the explanatory data analysis step provides little benefit, despite what I imagine is a large computational cost. Table 4 shows that the benefit of using such features in deep learning models is minimal.
- I feel that human in the loop feedback could benefit the method - allow a human to guide each iteration with insights gained from the automatic EDA. Additionally, insights from EDA are often not with the goal of improving classification, but understanding where the model will make mistakes or visualizing and comparing distributions of certain features.
- The claim that the model is more human-aligned than others is not proven, I believe human experiments would be necessary to say that this model is more human-aligned than others.
- I believe the evaluation focuses on the wrong aspect. In terms of EDA, if the authors could show that the data analysis produced is more human-like or more informative to humans than other methods, this would be a benefit of the proposed method.

**Questions:**

- What are the additional computational costs of each module in the method? I imagine that the cost is large compared to the very minimal performance gain.

---

> ### Author Response · Authors · 2025-11-20
> **1. Why Visual-Language Models? Justification Beyond Plot Generation**
>
> We appreciate the reviewer’s question regarding the necessity of incorporating a VLM. Our intention is not to just emulate “human-like EDA”, but to enhance the model’s ability to detect **statistical and structural patterns** that text-only LLMs often overlook when restricted to numeric summaries alone. The visual modality provides complementary information-particularly regarding distributional shapes, interactions, and localised irregularities is difficult to infer from tabular statistics, thereby enabling more informed and contextually grounded feature engineering decisions.
>
> Below, we clarify why visual EDA is complementary-and sometimes essential improving downstream feature engineering.
>
> ## **1. Statistical summaries alone are insufficient for LLM-driven EDA**
>  ----------------------------------------------------------------------
>
> When prompted for EDA, LLMs overwhelmingly default to:
> *   `df.describe()`, missing-value summaries
> *   value counts
> *   correlation matrices
>
> While useful, these miss three essential categories of information:
>
> ### **A. Distribution Geometry (shape, skew, modality, outliers)**
>
> A histogram or KDE encodes several statistical properties simultaneously (skewness, modality, kurtosis, tails, outliers).
> Reconstructing these from text summaries requires multiple separate metrics; LLMs frequently overlook or misinterpret them unless explicitly prompted.
>
> ### **B. Non-linear & multimodal relationships**
>
> Visualizations such as scatterplots, pairplots, and heatmaps reveal:
> *   heteroscedasticity
> *   cluster structure
> *   U-shaped or curved relations
> *   segmented population behaviour
>
> These patterns **do not appear** in correlation matrices.
> A correlation of ~0 can still hide a strong nonlinear signal visible immediately to a VLM.
>
> ### **C. Interaction effects across features**
>
> Plots like faceted histograms, pairplots, and 3D scatter plots surface **interaction-driven separability**, which is crucial for discovering derived features.
> This is precisely where VLMs excel: **visual geometry compresses high-dimensional statistical information into a single image**, which the model can decode more reliably than text-only summaries.
>
> * * *
>
> ## **2. Empirical evidence supports stronger reasoning with visual encodings**
> ---------------------------------------------------------------------------
>
> Multiple influential works demonstrate that VLMs outperform text-only LLMs on structure-understanding tasks:
> *   **PaLM-E (2023)** – vision inputs improve reasoning about structured environments
> *   **GPT-4V (2023)** – strong performance on chart interpretation and anomaly detection
> *   **ChartQA / PlotQA** – VLMs substantially outperform text-only models on chart reasoning
> *   **VisProg (2023)** – visual inputs enable better compositional reasoning
> *   **UniChart (2024)** – chart-based reasoning improves pattern detection tasks
>
> Across all these, the common conclusion is:
>
> > **Visual encodings carry richer structural patterns than raw numeric summaries, and models extract more reliable signals from them.**
>
> * * *
>
> ## **3. Visual EDA is not used in isolation-our system integrates both textual and visual signals**
> ------------------------------------------------------------------------------------------------
>
> Importantly, our method **does not ignore statistical EDA**.  We had added an additional section to our Appendix showcasing examples that highlight the interpretability aspect of the features from the EDA reports, refer `Appendix A.6 (EDA-DRIVEN INTERPRETABILITY: DATASET-SPECIFIC AGENT TRAJECTORIES EXAMPLES)`.
>
> *   few trajectory steps generate _no plots at all_
> *   numerical summaries (`describe`, correlations, missing-value patterns) are fully incorporated
> *   both plot outputs **and** textual summaries from executed code are fed back to the LLM
>
> Thus, the VLM is not a replacement for statistical tools but a **complementary encoder** when structural patterns are easier to read visually.
> The emphasis on plots stems from their ability to front-load high-value insights during early exploration, not from a rigid preference. Several trajectory steps in our experiments relied exclusively on statistical summaries because the agent determined that visual inspection was unnecessary.
>
> * * *
>
> ## **4. Why this matters for downstream ML tasks**
> -----------------------------------------------
>
> Our objective is not simply generating “plots”, but **discovering engineered features that improve downstream performance**.
> Visual patterns (clusters, multimodal behaviour, nonlinear trends, separability) directly motivate new features that purely numerical EDA frequently fails to reveal.
>
> Thus, the VLM is used because:
>
> *   it surfaces **additional structure** that text-only LLMs routinely miss
> *   this structure directly contributes to **higher-quality feature proposals**
> *   and these features demonstrably improve downstream model accuracy

---

> ### Author Response · Authors · 2025-11-20
> **2. Reassessing Human Alignment: Cost–Benefit Considerations and the Need for Human-Centered EDA Evaluation**
>
> ## **Performance Discussion in Context with EDA**
>
> We acknowledge that the performance gains reported in `Table 2` are modest when comparing the different modes of the PIFE framework. However, the additional cost introduced by employing a VLM for EDA is justified by the substantially richer and more interpretable set of analysis-grounded feature transformations it enables. Each generated feature is accompanied by a trace and an explicit reasoning chain explaining its construction. This information is comprehensively documented in the EDA reports `(Appendix A.6: EDA-Driven Interpretability-Dataset-Specific Agent Trajectory Examples)`, which illustrate the value added by integrating VLMs into the pipeline. These reports are human-readable, allowing data scientists to verify the rationale behind feature generation and to accelerate their understanding of the dataset.
>
> Moreover, the incremental computational cost is offset by the significant reduction in human effort that would otherwise be required to perform an equivalent EDA manually. This represents a major practical advantage.
>
> | Model          | EDA cost per dataset | FE cost per dataset | Total cost per dataset |
> | -------------- | -------------------- | ------------------- | ---------------------- |
> | PIFE (gpt-4.1) | $0.93894             | $0.26953            | $1.2085                |
> | PIFE (gpt-5)   | $4.1549              | $0.7905             | $4.9454                |
>
> * * *
>
> The results in Table 4 rely on RF feature importance scores, which are known to transfer poorly to MLPs or other deep architectures. However, when deep models themselves are integrated into the EDA-driven feature-generation loop (`Appendix 7: Extending PIFE to Deep Learning Models`, we observe consistent performance gains over standard deep learning baselines.
>
> Overall, our approach provides a practical framework for data scientists that combines automated reasoning, interpretability, and low operational cost, while delivering modest but reliable improvements in predictive performance.
>
> ## **Human Alignment and Analysis**
>
> We thank the reviewer for the thoughtful comments. We agree that demonstrating human-alignment rigorously requires controlled human studies, and we appreciate the suggestion. Our intent was not to claim global human alignment, but to show that the _style, depth, and coverage_ of the EDA performed by PIFE closely mirrors what experienced practitioners typically produce. To make this concrete, we benchmarked PIFE on several Kaggle datasets and directly compared its EDA outputs with **Kaggle Gold Medal notebooks**, which are widely regarded as high-quality, human-crafted analyses.
>
> For example, on the _Pima Indians Diabetes_ dataset, PIFE independently generated a range of analyses commonly found in top-ranked notebooks - such as stratified distribution checks, target correlation studies, outlier diagnostics, violin/box plots, scatter plots, and pairwise relationships - closely matching the insights seen in notebooks by human experts (e.g., msjahid’s [Gold Medal notebook](https://www.kaggle.com/code/msjahid/diabetes-risk-analysis-pima-indians-exploration)). Across multiple datasets, we observed that PIFE consistently surfaced the same categories of observations that human data scientists prioritize, and in some cases produced **more detailed, multi-step insights** that are absent from many human-written EDA notebooks.
>
> We fully agree that evaluating _EDA usefulness_ and _interpretability for humans_ is inherently subjective and benefits strongly from human-in-the-loop experiments. While our qualitative comparisons with expert notebooks offer an initial proxy for human-likeness, we see great value in designing formal human evaluations; such as rating studies or practitioner-level usability assessments-to strengthen this dimension. We appreciate the reviewer highlighting this, and we consider such human-centered evaluation an important direction for future work.
> Finally, our motivation is that EDA primarily serves **humans** in data science workflows. By producing analyses that reflect the structure and reasoning patterns observed in high-quality human EDA, PIFE aims to augment practitioners-either as a co-pilot or an end-to-end assistant, accelerating insight discovery and supporting more meaningful feature engineering. We believe that integrating application-specific models (e.g., VLMs for richer visual reasoning) will further enhance this human-facing alignment in future iterations.

---

> ### Author Response · Authors · 2025-11-20
> **3. Human-in-the-Loop & EDA Design Rationale**
>
> ###  **Part-1: Human in the Loop (HIL)**
>
> We thank the reviewer for highlighting the value of incorporating human-in-the-loop feedback. We fully agree that allowing human experts to guide iterations, especially by leveraging insights extracted during automatic EDA, could substantially enhance the quality and relevance of the generated features. As noted in `Section 5 (Conclusion)` of the paper, integrating HIL is a natural extension of our framework and an exciting future direction.
>
> For benchmarking, however, we intentionally run the system in a fully autonomous mode to ensure a fair and reproducible comparison with existing AutoFE baselines, which also operate without human guidance.
>
> ### **Part-2: Role of EDA in Our Framework**
>
> We appreciate the reviewer’s observation on the goal of exploratory data analysis. In our design, the EDA block is not instructed to directly optimize downstream predictive performance. Instead, its objective is to help the LLM build a deeper understanding of the dataset, capturing feature distributions, relationships, correlations, and structural patterns. These insights form the conceptual foundation of any feature engineering process.
>
> We intentionally avoid mixing EDA with performance-oriented instructions. LLMs are highly sensitive to prompt composition, and adding optimization intents alongside data understanding tasks can introduce ambiguity, disperse the model’s attention, and reduce controllability. Keeping the EDA block focused on pure data understanding helps produce clearer, more stable, and interpretable results, which the downstream feature-engineering agent then leverages.

---

### Official Review · Reviewer_mHis · 2025-10-30

**Soundness:** 4
**Presentation:** 3
**Contribution:** 4
**Rating:** 8
**Confidence:** 3

**Summary:**

This paper proposes PIFE (Progressive Insight Driven Feature Engineering), which selects features by iteratively doing EDA (exploratory data analysis), making plots and analyzing the plots with LLM and VLM and code execution. Their method takes advatange of human-like reasoning process where humans look at data analaysis plots to decide about the results. Each iteration, they use feature importance from the random forest to decide which features to keep. They are the first to propose automated feature engineering framework that integrates textual and visual exploratory data insights into a unified, iterative pipeline. They compared with a comprehensive set of baselines and show that their method selects better features. They also show the effectiveness of EDA in an ablation comparing with removing EDA from their pipeline. Although it doesn't help two datasets with EDA, it helps most. They also tried to further use OpenFE on the end outputting features from PIFE, and find there only to be a small improvement, showing the effectiveness of PIFE.

**Strengths:**

1. The paper proposes a very nice and intuitive method (using insights from data exploratory phrase, and using an VLM to achieve that) to improve auto feature engineering.
2. The results show that PIFE is better than other methods.
3. The paper has comprehensive results to isolate effects of EDA and also how combining their method with OpenFE works.

**Weaknesses:**

1. The method replies on the model being an inherently interpretable model that exposes feature importance, which would not be true for deep learning models, although we could use some post-hoc methods to approximate the feature importance.

**Questions:**

1. How would the proposed method work if the underlying predictive model is not random forest but a neural network?

---

> ### Author Response · Authors · 2025-11-20
> **Neural Network Predictive Model**
>
> Thank you for the insightful question. To evaluate the generality of our approach beyond tree-based models, we applied PIFE using fully-connected MLPs and TabPFN as the underlying predictive models, instead of Random Forests. The results demonstrate that the proposed framework is model-agnostic and continues to provide consistent improvements even when the predictor is a neural network.
>
> We ran the full PIFE pipeline with:
> - MLP classifiers/regressors (parameters are in `Table 13`, Feature importance design in `Appendix A.7.1` of the paper)
> - TabPFN, a transformer-based neural predictive model
>
> On six datasets spanning both regression and classification tasks, the PIFE-enhanced models consistently improved over their neural baselines:
>
> > **Table: PIFE performance with MLP as predictive model**
> | Dataset         | Baseline      | PIFE (Ours)       | % Improvement |
> | --------------- | ------------- | ----------------- | ------------- |
> | pima_indian*    | 0.720 ± 0.008 | **0.731 ± 0.006** | 1.528         |
> | fertility*      | 0.840 ± 0.00     | **0.857 ± 0.025** | 2.024         |
> | housing_boston† | 0.678 ± 0.002 | **0.694 ± 0.006** | 2.36          |
> | airfoil†        | 0.728 ± 0.003 | **0.780 ± 0.003** | 7.143         |
> | openml_586†     | 0.582 ± 0.006 | **0.771 ± 0.024** | 32.474        |
> | adult*          | 0.851 ± 0.002 | **0.852 ± 0.001** | 0.118         |
>
> With MLP models, PIFE improved performance on all datasets, with gains as high as +32.5% on openml_586 and +7.1% on airfoil.
>
> > **Table: PIFE performance with TabPFN as predictive model**
> | Dataset         | Baseline          | PIFE (Ours)       | % Improvement |
> | --------------- | ----------------- | ----------------- | ------------- |
> | pima_indian*    | 0.760 ± 0.005     | **0.768 ± 0.008** | 1.053         |
> | fertility*      | 0.880± 0.000         | **0.880± 0.000**         | 0             |
> | housing_boston† | 0.736 ± 0.002     | **0.737 ± 0.005** | 0.136         |
> | airfoil†        | **0.886 ± 0.002** | 0.884 ± 0.002     | -0.226        |
> | openml_586†     | 0.855 ± 0.002     | **0.857 ± 0.001** | 0.234         |
>
> With TabPFN, PIFE also yielded performance improvements on most datasets (e.g., +0.234% on openml_586 and +1.05% on pima_indian), demonstrating that the gains are not restricted to tree-based predictors.
>
> These results indicate that:
>
> - PIFE integrates seamlessly with neural models, requiring no architectural modifications.
>
> - The benefit does not depend on the model class, confirming that the proposed feature-enhancement process is compatible with both classical machine learning and modern neural architectures.
>
> - The extracted feature importances (`Appendix A.7.1`) remain meaningful when computed from neural predictors, and the generated explanations help improve downstream model accuracy even without tree-structured models.

---

### Official Review · Reviewer_2ZXd · 2025-11-01

**Soundness:** 4
**Presentation:** 3
**Contribution:** 3
**Rating:** 6
**Confidence:** 4

**Summary:**

The paper presents PIFE, an AutoFE framework that leverages multimodal reasoning (LLMs + VLMs) to automate EDA, insight extraction, and feature generation for tabular datasets. PIFE iteratively generates statistical summaries and visualizations, interprets them, and synthesizes candidate features as executable Python code. Downstream model feedback is used to refine feature generation. Experiments across 22 datasets show PIFE outperforms existing AutoFE baselines in interpretability and context-awareness.

**Strengths:**

Novelty: First to tightly integrate textual and visual EDA insights in an iterative, feedback-driven AutoFE pipeline.
Interpretability: Features are generated as symbolic programs, ensuring transparency and reproducibility.
Empirical Rigor: Evaluated on diverse datasets, with fair comparisons and ablation studies.
Performance: Consistently outperforms baselines in both classification and regression tasks.
Generalization: Features transfer well to different downstream models and unseen datasets.
Open Science: Code and datasets are released for reproducibility.

**Weaknesses:**

Scope and Robustness of EDA
- The EDA routines in PIFE are primarily statistical and visual, focusing on distributions, correlations, and categorical/temporal trends. This scope is well-suited for standard tabular data, but may not generalize to domains requiring specialized EDA (e.g., time series, graphs, or highly unstructured data).
- Robustness of the approach depends on the diversity and quality of EDA routines. If the EDA is limited or misses important patterns, the generated features may be suboptimal.
- The paper notes that in some datasets, EDA-driven features do not improve performance, especially when original features are already strong or datasets are small.

Supported Visualizations
- PIFE supports common tabular visualizations: histograms, scatter plots, heatmaps, LOWESS curves, binned means, and categorical plots (bar, boxplots).
- The framework does not appear to support more advanced or domain-specific visualizations (e.g., time series plots, network diagrams).
- The quality and diversity of insights are limited by the types of visualizations produced and interpreted.

LLM Hallucinations and Reliability
- LLMs may generate plausible but incorrect or ungrounded features, especially if the EDA context is noisy or incomplete.
- The framework mitigates hallucinations via downstream validation (feature importance feedback, cross-validation, feature selection), but does not provide explicit mechanisms for detecting or correcting hallucinations at the insight or feature generation stage.
- No formal guarantees are provided against hallucinations; robustness is empirically assessed via multiple seeds and ablation studies.
- There is a risk of overfitting to spurious patterns or memorized solutions from prior competitions.

**Questions:**

See weaknesses

---

> ### Author Response · Authors · 2025-11-19
> **Clarifications on EDA Scope, Visualization Coverage, and LLM Reliability in PIFE**
>
> We appreciate the reviewer’s insights, and the added experiments and clarifications directly address the concerns about robustness and generality.
>
> ## EDA Scope and Robustness
>
> **1. Generalization to Time-Series Data (New Experiments Added)**
>
> While the original submission focused on tabular EDA, PIFE already includes temporal diagnostics such as autocorrelation, seasonality checks, lag-based summaries, rolling-window statistics, and decomposition plots. To validate this, we added **new benchmarks on five time-series competitions** added to `Appendix A.4: ADDITIONAL EXPERIMENTS AND EDA-GUIDED FEATURE EXAMPLES`. As shown in the below table, PIFE improves over strong baselines on most datasets-for example **+13.4% on Beef**, **+4.8% on Coffee**, and **+4.4% on ItalyPowerDemand**-demonstrating that EDA-driven feature synthesis meaningfully helps in temporal domains as well.
>
> > **Table: PIFE performance on time series classification datasets from the UCR Time Series Archive using gpt-5 as the LLM and VLM.**
>
> | Competition | Overall Baseline | Overall PIFE | % improvement |
> |-------------------|------------------|---------------|----------------|
> | italypowerdemand | 0.67 ± 0.06 | 0.70 ± 0.05 | 4.47761194 |
> | gunpoint | 0.99 ± 0.01 | 1.00 ± 0.00 | 1.01010101 |
> | coffee | 0.83 ± 0.02 | 0.87 ± 0.02 | 4.819277108 |
> | ecg200 | 0.96 ± 0.01 | 0.99 ± 0.00 | 3.125 |
> | beef | 0.82 ± 0.01 | 0.93 ± 0.02 | 13.41463415 |
>
> **2. Features and Supporting EDA Visualiations**
>
> For better understanding of the impact of EDA and how generated features are grounded by much complex and detailed analysis and statistics; we present a few example features and underlying EDA analysis used to build the features in `Appendix A.4` of the paper.
>
> **3. Dependence on EDA Diversity**
>
> PIFE is designed as a **modular AutoFE pipeline**, enabling seamless integration of new data types, models, and evaluation metrics. Extending to time-series required _minimal code changes (editing respective function files as seen in the code provided) and no prompt changes_, reflecting the flexibility of the architecture.
>
> That said, **AutoFE for graphs and unstructured modalities is fundamentally different**-these domains rely on learned representations or topology-aware modeling rather than explicit composable transformations. As they require distinct architectures and search spaces, they fall outside the scope of this work, which intentionally focuses on structured/tabular and time-series data.
>
> **4. Datasets with Limited Improvement**
>
> We acknowledge that improvements may be minimal in small or already feature-rich datasets. This clarification has been added, while the new time-series results illustrate cases where PIFE provides notably larger gains.
>
> ## Supported Visualisations
>
> We thank the reviewer for the thoughtful feedback. PIFE’s EDA pipeline is organized into three hierarchical stages (as shown in `Appendix A.9 Listing 7: EDA Analysis Code Generation` check Focus Areas); from distribution and correlation checks, to interaction and non-linear pattern discovery, and finally to temporal, categorical, and outlier-focused analysis. This staged design already accommodates both tabular and time-series visualizations, and it enables the agent to progressively expand its analytical depth.
>
> We agree that incorporating more advanced or domain-specific visualizations would further strengthen the system’s ability to surface richer insights. The hierarchical structure provides a clear pathway for extending PIFE with additional visualization types and deeper modality-specific routines in future versions
>
> ## Hallucinations and Reliability
>
> PIFE mitigates hallucinations by grounding feature generation in data-derived EDA insights; ensuring features reflect actual statistical and visual patterns rather than domain priors. While VLMs themselves may misinterpret plots, we can benchmark them on chart understanding datasets (e.g., [ChartQA](https://arxiv.org/abs/2203.10244), [ChartQA Pro](https://arxiv.org/abs/2504.05506)) to quantify and improve reliability. Combined with downstream validation (cross-validation, feature importance pruning), this provides a practical defense against ungrounded or spurious features.

---

> ### Comment · Reviewer_2ZXd · 2025-11-27
>
> Thanks for the additional experiments and the details response - they clarify my concerns.

---

### Official Review · Reviewer_zXFD · 2025-11-04

**Soundness:** 2
**Presentation:** 1
**Contribution:** 2
**Rating:** 4
**Confidence:** 4

**Summary:**

This paper introduces an AutoFE framework that incorporates automatic data analysis like EDA, leveraging LLMs and VLMs. A VLM generates statistical summaries of features from EDA plots that are used as the context in the LLM-based AutoFE process. The paper also explores the effects of different feature selection methods.

**Strengths:**

This paper suggests a good direction of incorporating the visual information of datasets in AutoFE. The paper presents comprehensive experimental results involving both traditional and deep learning downstream models. The experimental setup is explained clearly.

**Weaknesses:**

While it is beneficial to enrich the context information of LLM-based AutoFE, I do not think the automatic data analysis process has been well integrated. Algorithm 1 suggests that the process starts from scratch at each iteration, which is quite inefficient. The data analysis process also seems quite random, and there is no feedback mechanism to guide it. From the ablation study, the performance gain is limited.

The presentation of the paper is not very clear in some parts especially Section 3. More detailed explanations may help. Algorithm 1 is a bit hard to follow and not positioned appropriately.

In the experimental section, the statistical significance of results is not reported. It would be great to also include a cost study of the framework.

**Questions:**

Does the framework represent feature transformations in code or RPN? Listing 3 seems to suggest that both are adopted. I think this is unnecessary and may create inconsistencies.

What VLM has been used in experiments?

In A.5.2, why are the parameters presented as ranges, different from A.5.1?

Somehow the repository shows “the requested file is not found”  and the code is inaccessible.

---

> ### Author Response · Authors · 2025-11-20
> **Assessment of Iterative EDA: Efficiency, Guidance, and Effectiveness**
>
> ## Justification on Efficiency of EDA
>
> Features generated in earlier iterations are intentionally reused during later iterations, as reflected by the ~57% reuse rate shown in the Table below. Because these newly created features become part of the evolving dataset, it is essential to re-run EDA at each iteration to understand their interactions with existing features. While this can introduce some redundancy, many analyses (e.g., correlation or interaction analysis) must be repeated in context, since new features can significantly alter statistical relationships and downstream insight quality.
>
> Each iteration, consisting of a 3-stage hierarchical EDA cycle followed by feature generation, is self-contained. As the feature space grows, the distributional properties and dependency structures of the dataset change accordingly. Performing EDA at every iteration is therefore necessary, not optional, to ensure that insights remain accurate, relevant, and reflective of the transformed dataset, ultimately enabling meaningful and well-grounded feature construction.
>
> > **Table:** Shows the frequency of newly generated features being used to create further new features
>
> | Category | Freq. of Event | Total Features Generated |
> | --- | --- | --- |
> | Newly generated features reused across iterations | 5541 | 9682 |
>
> It is a valid point that across the set of various datasets spanning various domains, the overall performance gain is limited. However, taking a closer look at what our framework achieves is important to understand the merits of using visual clues through EDA to make more grounded decisions regarding the feature engineering problem. From our extensive experimentation, a few interesting examples can help in getting a better idea of the overall benefit, interpretability validation, and reason behind each feature generated compared to older methods on pure randomness and brute force. (Check `Appendix A.6` for examples.)
>
> ## Hierarchical Structure of EDA
>
> We thank the reviewer for the thoughtful observation. While the full logic is abstracted into a helper function, Appendix A.12 (Listing 1: _EDA Analysis Code Generation_) outlines how our data analysis is governed by a **three-stage hierarchical EDA strategy**, rather than random exploration. Each stage has clearly defined focus areas - ranging from distribution and correlation analysis, to interaction and non-linear pattern discovery, and finally to temporal/categorical structure and outlier analysis. This hierarchy allows the agent to **incrementally build analyses**, where each stage informs the next, providing an implicit feedback mechanism.
>
> The framework delivers **state-of-the-art overall results among peer AutoFE systems** while offering significantly **higher interpretability**, owing to its EDA-driven reasoning process. To the best of our knowledge, PIFE is also **the first framework to integrate hierarchical, human-style EDA directly into the AutoFE pipeline**, which we believe is a key conceptual contribution. We appreciate the reviewer raising this point and agree that extending the feedback loop further is a promising direction for future work.
>
> ## Regarding Presentation of `Section-3: Methodology` and Algorithm Complexity
>
> We appreciate the reviewer pointing out the complexity of some parts of the paper. We have revised the `Section-3: Methodology` and `Algorithm-1` in the main paper with simpler flow and more detailed explanations of various components.
>
> ## Statistical Significance and Cost Analysis
>
> We have added the analysis of cost in `Appendix A.14` for reference.
>
> > Statistical Significance Across Methods
>
> | Comparison             | t-statistic | p-value       | Interpretation                                         |
> | ---------------------- | ----------- | ------------- | ------------------------------------------------------ |
> | CAAFE vs OCTree | 3.6698  | 0.0009    | CAAFE performs **significantly better** than OCTree |
> | CAAFE vs PIFE | –2.4065 | 0.0216    | PIFE performs **significantly better** than CAAFE |
> | OCTree vs PIFE | –5.0073 | 1.10×10⁻⁵ | PIFE performs **significantly better** than OCTree |
>
> > Summary
>
> - PIFE is the best overall (statistically significantly higher mean scores than both CAAFE and OCTree).
> - CAAFE is better than OCTree, but worse than PIFE.
> - OCTree is the weakest among the three.
>
> ## Response to the Questions
>
> - We have included the explanation for this in the `Section-3 Methodology`, and  `Appendix A.8` also talks about the scope of use of RPN. We primarily use RPN to perform post-hoc feature order analysis.
> - We have used gpt-4.1 and gpt-5 as the VLMs and LLMs, both due to their exceptional vision analysis capabilities.
> - We have corrected the hyperparameters as can be seen in `Appendix A.10.2` to fixed values.
> - Repository link was somehow corrupted, and we have updated the link in `Appendix A.1`.

---

### Meta-Review · Area_Chair_iiCP · 2026-01-07

**Summary:**

this paper received scores of 4,4,6,8. the reviewers who voted to reject highlighted the following concerns:

Concerns with the algorithm, inefficiency of Algorithm 1 starting from scratch at each iteration, necessity of a VLM.

Presentation of the paper is not very clear in some parts especially Section 3 and Algorithm 1.

Experimental concerns: lack of statistical significance of results, cost study of the framework, that the explanatory data analysis step provides little benefit despite large computational cost, and the benefit of using such features in deep learning models is minimal.

Significant issues with human evaluation: human in the loop feedback could benefit, the claim that the model is more human-aligned than others is not proven, for EDA authors should show that the data analysis produced is more human-like or more informative to humans than other methods.

**Reviewer Concerns:**

Concerns with the algorithm, inefficiency of Algorithm 1 starting from scratch at each iteration, necessity of a VLM.

--> not really addressed, the algorithm does end up being slower, with limited performance improvement.

Presentation of the paper is not very clear in some parts especially Section 3 and Algorithm 1.

--> subjective, but i feel it is ok.

Experimental concerns, lack of statistical significance of results, ablation studies show limited improvement of key components.

--> in my opinion, performance gains do seem to be small despite increased computational cost.

Lack of human evaluation: no human in the loop feedback, no human judgement of interpretability.

--> biggest concern in my opinion, and not addressed. without proper user studies or human in the loop eval there is no justification for improved interpretability. this, coupled with increased complexity and limited performance improvements, limits the impact of the method.

**Reviewer Scores:**

- 96TH who gave 4 unlikely to increase their score.

- zXFD who gave a 4 might have increased to 6 with 50% chance, 50% same score.

---

### Decision · Program_Chairs · 2026-01-26

Reject